# Broadly neutralizing and protective nanobodies against SARS-CoV-2 Omicron subvariants BA.1, BA.2, and BA.4/5 and diverse sarbecoviruses

Mingxi Li[1,15], Yifei Ren[2,3,15], Zhen Qin Aw [4,5,6,15], Bo Chen[7,15], Ziqing Yang[1], Yuqing Lei[1], Lin Cheng[8,9], Qingtai Liang[1], Junxian Hong [1], Yiling Yang [1], Jing Chen[2,3], Yi Hao Wong [4,5,6], Jing Wei[1], Sisi Shan[1], Senyan Zhang [2], Jiwan Ge [2,3], Ruoke Wang [1], Jay Zengjun Dong[10], Yuxing Chen[11], Xuanling Shi[1], Qi Zhang[1], Zheng Zhang [8,9], Justin Jang Hann Chu [4,5,6,12] ✉, Xinquan Wang[2] ✉ & Linqi Zhang [1,13,14] ✉

As SARS-CoV-2 Omicron and other variants of concern (VOCs) continue spreading worldwide, development of antibodies and vaccines to confer broad and protective activity is a global priority. Here, we report on the identification of a special group of nanobodies from immunized alpaca with potency against diverse VOCs including Omicron subvariants BA.1, BA.2 and BA.4/5, SARS-CoV-1, and major sarbecoviruses. Crystal structure analysis of one representative nanobody, 3-2A2-4, discovers a highly conserved epitope located between the cryptic and the outer face of the receptor binding domain (RBD), distinctive from the receptor ACE2 binding site. Cryo-EM and biochemical evaluation reveal that 3-2A2-4 interferes structural alteration of RBD required for ACE2 binding. Passive delivery of 3-2A2-4 protects K18-hACE2 mice from infection of authentic SARS-CoV-2 Delta and Omicron. Identification of these unique nanobodies will inform the development of next generation antibody therapies and design of pan-sarbecovirus vaccines.

As the severe acute respiratory syndrome coronavirus 2 (SARS-CoV-2) continues to rage globally, we have been witnessing the rapid emergence and turnover of multiple variants of concerns (VOCs) such as Alpha (B.1.1.7) initially found in the United Kingdom; Beta (B.1.351) in South Africa; Gamma (P.1) in Brazil; Delta (B.1.617.2) in India; and Omicron subvariants in Botswana and South Africa (https://www.who.int/en/activities/tracking-SARS-CoV-2-variants/). These VOCs are not only associated with steeply increased new infections among unvaccinated but also break-through infections among the infected and vaccinated individuals[1–4]. Increasing evidence suggests that substantial changes in their antigenic properties have facilitated these VOCs to escape from serum neutralization of convalescent and vaccinated

individuals[5–9]. As a result, efficacies of all vaccine modalities as well as many therapeutic antibodies approved for emergency use authorization (EUA) have been severely compromised, particularly toward Omicron subvariants BA.1, BA.2, BA.3, and BA.4/5, followed by Beta, Delta, Gamma, and to the least extent by Alpha[7,10–12]. Omicron subvariants are perhaps the most insidious as they generally cause milder symptoms but carries the exceptionally high viral load in the upper respiratory tract with extremely high efficiency in transmission[13–15]. As quiet as it seems, Omicron subvariant BA.1, then BA.2, and now BA.4/5 have been actively replacing other VOCs and local variants to become the most dominant variant in many parts of the world. Development of broader and more effective therapies and vaccines against these

Omicron subvariants has therefore become an urgent and global priority.

One striking aspect of Omicron is the largest number of mutations found in the spike (S) protein among the VOCs identified thus far (https://www.gisaid.org), although the origins and mechanism of their accumulations remain unclear[16,17]. At least 35 substitutions were found in the S protein of Omicron compared to the prototype strain from Wuhan, China. Of which, about 15 are located in the RBD and 8 in the N-terminal domain (NTD), although the exact number of substitutions vary among different subvariants (https://www.gisaid.org). BA.1 and BA.2 are two early subvariants of Omicron that emerged around the end of 2021 and have since then been rapidly spreading worldwide. However, BA.4 and BA.5 subvariants, found in early April 2022 in Gauteng of South Africa, are actively replacing BA.1 and BA.2 and fueling the current wave of new and breakthrough infections in many parts of the world[18]. Genetically, BA.1 is rather unique while BA.2 (and its 12th and 75th lineage BA.2.12.1 and BA.2.75, respectively), BA.4, and BA.5 are highly related[19]. BA.3 appeared to be the mosaic between BA.1 and BA.2 (https://www.gisaid.org). BA.4 and BA.5 are believed to have evolved from BA.2 and share identical S sequences, thus frequently referred to as BA.4/5.

Recently, several elegant studies have pinpointed a few key substitutions in the S protein that are responsible for neutralization escape, and many of which are shared among Omicron subvariants or with other VOCs. For example, all Omicron subvariants have four substitution sites in common in the cryptic inner face of RBD, namely G339D, S371L/F, S373P, and S375F, and the last three are able to markedly reduce neutralization of many antibodies as they changed the conformation and biochemical properties of their residing loop[20]. In addition, the N501Y substitution previously shown to enhance binding affinity to the receptor angiotensin-converting enzyme 2 (ACE2) and found in Alpha, Beta, Gamma is also present in all Omicron subvariants[7,10,20,21]. BA.4/5, Delta, Epsilon, Lambda, and Kappa have substitutions L452R or L452Q within the RBD that facilitate virus escape[22–24]. Beta, Gamma, and Omicron each have three common substitution sites within the RBD, namely K417N/T, E484K/A, and N501Y, which resulted in marked reduction or complete loss of neutralizing activities of many therapeutic antibodies and immune serum from vaccinated individuals[5,10,25–27]. In the NTD, Alpha, Beta, Gamma, and Omicron share deletions/insertions and substitutions within or near the "NTD supersite" that largely consisted of the N1 (residues 14–26), N3 (residues 141–156), and N5 (residues 246–260) loops[28–31]. While these findings have clearly identified critical substitutions that confer viral escape from antibody neutralization, they also point to the very substitutions that have to be overcome for the broad and potent antibody therapeutics and vaccines.

To search for broadly neutralizing antibodies, we decided to use immunized alpaca with RBD and spike protein of SARS-CoV-2, as this animal species, like those in the family of *Camelidae*, encode a small single heavy chain nanobody (<15 kDa) with one variable domain (VHH) that are able to reach regions otherwise inaccessible by conventional human antibodies[32]. Their special properties in specificity, stability, thermotolerance, low immunogenicity, and ease in production in yeast and other cost-effective systems have made the nanobodies a desirable candidate for next-generation interventions against SARS-CoV-2 infection[33–47].

In this work, we identify a total of 593 nanobodies capable of binding to the recombinant spike trimer of prototype SARS-CoV-2 through screening nanobody library from an immunized alpaca. Of which, we discover a small but unique group of nanobodies with potency against all VOCs tested, including Omicron subvariants BA.1, BA.2, BA.2.12.1, and BA.4/5, as well as SARS-CoV-1 and diverse sarbecoviruses from bats and pangolins. The IC50 against a panel of 20 tested coronaviruses reaches as low as single or less nanomolar concentration, representing the most broad and potent nanobodies described to date. Passive delivery of one of the representative nanobodies, 3-2A2-4, protects K18-hACE2 mice from infection of authentic SARS-CoV-2 Delta and Omicron. Structural analysis reveals 3-2A2-4 targets to a unique and highly conserved epitope between the cryptic and outer face of RBD, distinctive from the ACE2 binding site. 3-2A2-4 appears to exert its neutralizing activity through affecting transitioning of RBD from its "down" to "up" conformation, a structurally required configuration from ACE2 binding. These results clearly indicate that we have identified a broadly neutralizing and protective nanobody that recognizes a highly conserved epitope among diverse SARS-CoV-1 variants, SARS-CoV-1, and sarbecoviruses. The nanobody and the epitope identified should provide an invaluable reference for the development of next-generation antibody therapies and vaccines against wide varieties of SARS-CoV-2 infection and beyond.

## Results

### Cross-neutralizing SARS-CoV-2 and SARS-CoV-1 nanobodies

We first constructed yeast VHH library based on the cDNA of peripheral blood lymphocytes from a sequentially immunized alpaca seven days after the last immunization. The immunization regimen included three-time subcutaneous injections of 200 μg recombinant RBD of prototype SARS-CoV-2 in Freund adjuvant, one-time subcutaneous injection of $10^{11}$ viral particle AdC68-19S vaccine expressing the prototype S trimer[48], and two-time subcutaneous injections of 200 μg recombinant S-2P protein of prototype SARS-CoV-2 in Freund adjuvant (Fig. S1a). We started with selection for cross-neutralizing SARS-CoV-2 and SARS-CoV-1 nanobodies, as these nanobodies would be expected to have a higher probability to cross-react with Omicron subvariants and other members of sarbecoviruses. Through the iterative process of FACS-sorting, enriching, and finally expressing in the recombinant form with human IgG1 Fc fragment, we identified a total of 593 nanobodies capable of binding to the recombinant spike trimer of prototype SARS-CoV-2. Of which, 124 showed neutralizing activity to prototype SARS-CoV-2 pseudovirus with IC50 ranging from 0.003 to 5.399 μg/mL or 0.039 to 71.039 nM (Fig. 1). Among these, 91 demonstrated cross-neutralizing activity to SARS-CoV-1, and all but two (No. 43 and 55) strongly bound to RBD (Fig. 1). Phylogenetically, these cross-neutralizing nanobodies were segregated into four major clusters (a, b, c, and d) (Fig. 1, red in the left panel). Clusters a and d nanobodies had equivalent average IC50 to SARS-CoV-2 and SARS-CoV-1 (0.077 vs. 0.056 μg/mL or 1.013 vs. 0.737 nM) while clusters b and c, however, demonstrated stronger activity to SARS-CoV-1 than to SARS-CoV-2 (0.036 vs. 0.499 μg/mL or 0.474 vs. 6.566 nM). This suggests that the epitopes recognized by clusters b and c nanobodies could be more exposed and/or easily accessible on the SARS-CoV-1 spike. Similar findings have recently been reported for human antibodies isolated from convalescent COVID-19 patients as well as COVID-19 mRNA vaccinees[49,50].

Genetic analysis of the 91 cross-neutralizing nanobodies identified preferred usage of several germline V-gene and J-gene, particularly for IGHV3S65 (39.6%), IGHV3-3 (37.4%), IGHV3S53 (13.2%), and IGHJ4 (100%) (Fig. S1b). Similar pattern of preference was also noticed among the total of 124 isolated nanobodies (Fig. S1c), and the 237 published nanobodies in the CoV-AbDab database (http://opig.stats.ox.ac.uk/webapps/covabdab/) except for IGHV3S65 (Fig. S1d). Such convergence on germline gene usage may suggest that the combinations of the gene segments encode nanobodies with unique structural and biochemical properties rendering them naturally complementary in shape and strong in binding to the spike surface of SARS-CoV-2 and SARS-CoV-1. Furthermore, the 91 cross-neutralizing nanobodies were all dominated by 19-residue long CDR3 (Fig. S1e and S1f) while those in the CoV-AbDab database by 17- or longer segments (Fig. S1g). However, the CDR3 length per se was unlikely a prerequisite for neutralization breadth across SARS-CoV-2 and SARS-CoV-1. Nanobodies in clusters *a* and *d* had an average CDR3 length of 10 and 19 residues,

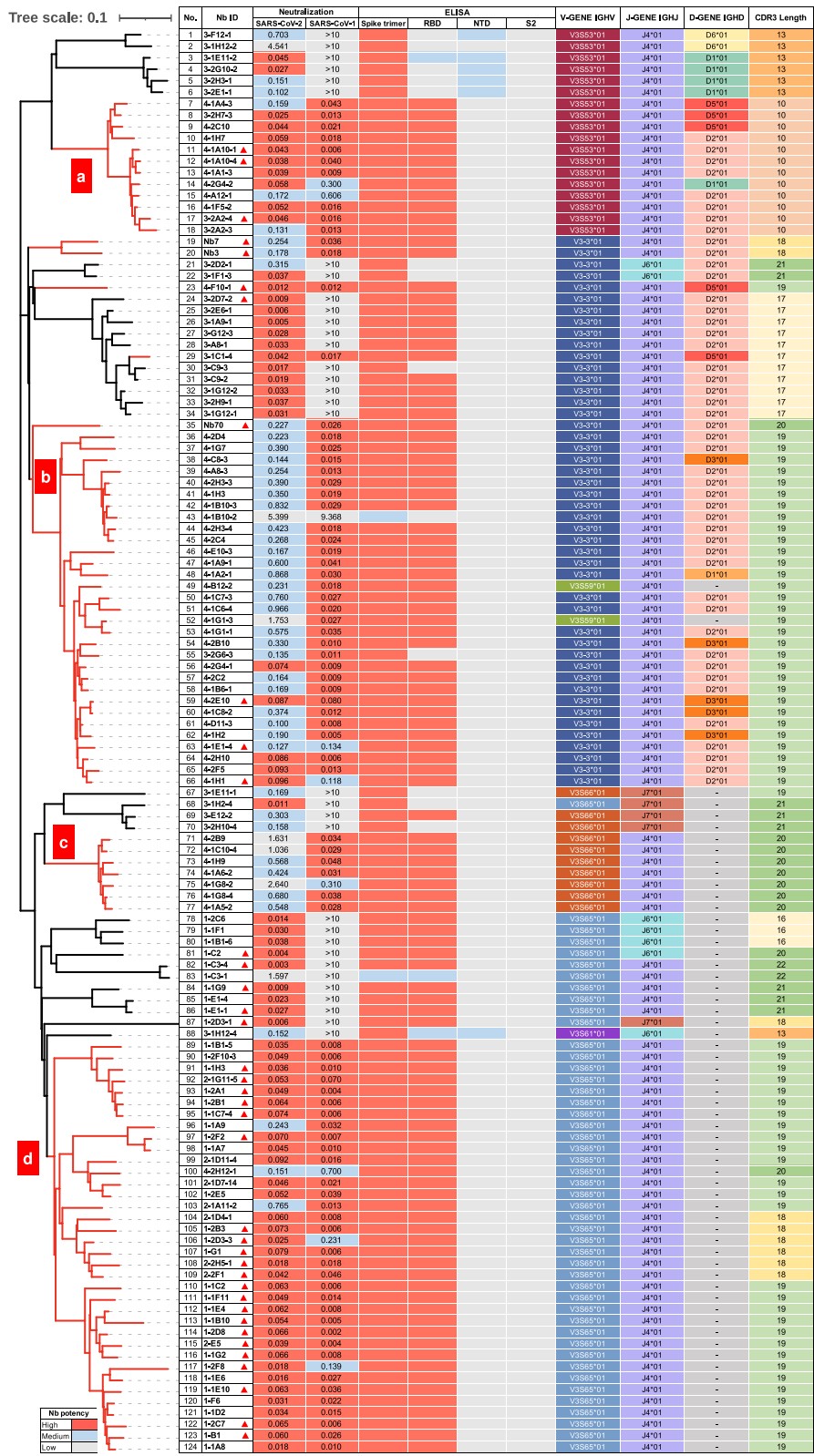

respectively, but demonstrated equivalent cross-neutralizing activity against SARS-CoV-2 and SARS-CoV-1 (Fig. 1). Interestingly, sequence logo plots identified 7 polar residues (GSYYYCS) in the 19-residue CDR3 that were rather prevalent among the 91 cross-neutralizing nanobodies and the 237 published nanobodies in the CoV-AbDab database, although no obvious patterns were found in 17- or 18-residue CDR3 sequences (Fig. S1h, S1i, and S1j).

## Broadly neutralizing nanobodies against Omicron subvariants and diverse sarbecoviruses

Next, we selected 32 cross-neutralizing and 6 SARS-CoV-2 specific nanobodies for in-depth analysis, based on their representation in neutralizing potency and locations on the phylogenetic tree and highlighted by the red triangles in Fig. 1. We first classified the 32 cross-neutralizing nanobodies into 3 groups based on their degree of

**Fig. 1 | Phylogenetic, neutralizing, binding, and genetic properties of isolated nanobodies against SARS-CoV-2 and SARS-CoV-1.** Phylogenetic analysis of 124 neutralizing nanobodies out of 593 binding nanobodies to the spike trimer of prototype SARS-CoV-2, isolated from the VHH library of an immunized alpaca. All nanobodies were constructed in Fc format and their cross-neutralization against SARS-CoV-2 and SARS-CoV-1 can be clustered in four groups (**a**, **b**, **c**, and **d**) and highlighted by the red branches. Neutralizing activities of each nanobody to SARS-CoV-2 and SARS-CoV-1 pseudoviruses are shown by half-maximal inhibitory concentration IC50 (μg/mL). The thirty-eight representative nanobodies selected for further evaluation are indicated by red triangle in Nb ID column. Results were calculated from three independent experiments and each performed in technical duplicates. The binding activity of each nanobody to the spike trimer, RBD, NTD, and S2 regions of prototype SARS-CoV-2 are indicated by the variable color scheme, with red for high binding (OD450 > 2), blue for medium binding (2 > OD450 > 1), and grey for low binding (OD450 < 1). The genetic features such as germline variable gene segment (V), diversity gene segment (D), and junction gene (J) as well as CDR3 length of each nanobody are indicated with various colors. See also Fig. S1. Source data are provided as a Source Data file.

competition with ACE2 and eCR3022, a previously published antibody with a known epitope on the cryptic face of RBD that only became accessible when RBD in the "up" standing conformation[51]. The Group 1 (G1) nanobodies strongly competed with both ACE2 and eCR3022 for binding to RBD (Fig. 2a), suggesting they also bound to similar cryptic epitopes on RBD from the angles that restricted ACE2 binding. The Group 2 (G2) nanobodies moderately competed with ACE2 but strongly with eCR3022, indicating similar cryptic epitopes like those in G1 but less interfering with ACE2 binding. The Group 3 (G3) nanobodies, however, minimally competed with ACE2 or eCR3022, suggesting their binding poses deviated away from that of ACE2 and eCR3022. The Group 4 (G4) nanobodies included the 6 SARS-CoV-2-specific nanobodies for control purpose, and showed varying competition activity with ACE2 and the weakest with eCR3022 among all nanobodies studied here (Fig. 2a).

We next studied the neutralizing breadth of these nanobodies against a panel of 19 sarbecoviruses that were dependent on ACE2 for entry, including 14 major SARS-CoV-2 variants and subvariants, SARS-CoV-1, and 4 representatives from bats and pangolins. The G1 nanobodies showed broad neutralizing activity against all viruses tested except for Omicron BA.1, BA.2, BA.2.12.1, and BA.4/5 where their activity dropped below the detection limit (BDL in dark red) (Fig. 2a, Fig. S2 and Table S1). Interestingly, many nanobodies showed improved neutralizing activity to SARS-CoV-1, pangolin coronavirus GD, and bat coronavirus RaTG13 (in blue) while varied considerably to pangolin coronavirus GX and bat coronavirus WIV16 (mixed with red and blue) (Fig. 2a, and Fig. S3). Similar findings have also been reported for human antibodies isolated from convalescent COVID-19 patients as well as COVID-19 mRNA vaccinees[49]. The G2 nanobodies appeared to be less affected by Omicron subvariants. In particular, Nb70 could neutralize BA,1, BA.2.12.1, and BA.4/5 with similar potency to WT D614G, although about 13.8-fold reduction against BA.2. Like those in the G1, the G2 nanobodies also demonstrated improved neutralizing activity to SARS-CoV-1, and those representative coronaviruses from bats and pangolins (Fig. 2a). Remarkably, the G3 nanobodies exhibited the broadest neutralizing activity against all viruses tested including Omicron subvariants, although the moderate or substantial reduction was found to pangolin coronavirus GX. The G4 nanobodies, despite being the strongest against WT D614G with an average IC50 0.014 μg/mL or 0.184 nM, had the poorest breadth against the 19 viruses tested. Severe reduction or complete loss of neutralizing activities were found in many nanobodies with the only exception to Alpha, Epsilon, and A23.1 variants and pangolin coronavirus GD. It was likely that the G4 nanobodies targeted sites either overlapped with or proximate to receptor-binding motif (RBM), consistent with their notable competition with ACE2 for binding to RBD (Fig. 2a).

We then selected one nanobody each from G1 (1-2C7), G2 (Nb70), and G3 (3-2A2-4) to test their neutralizing activity against authentic SARS-CoV-2 including the wildtype (WT), Alpha, Beta, Delta and Omicron BA.1 subvariant, using focus reduction neutralization test (FRNT). Consistent with their respective activities to pseudoviruses, 3-2A2-4 from G3 was the most broad and potent among the three

nanobodies with IC50 of 0.102 μg/mL against WT, 0.115 μg/mL against Alpha, 0.098 μg/mL against Beta, 0.130 μg/mL against Delta, and 0.106 μg/mL against Omicron BA.1 (Fig. 2b). Nb70 had a relatively moderate IC50 values with 1.337 μg/mL against WT, 1.242 μg/mL against Alpha, 1.635 μg/mL against Beta, 1.210 μg/mL against Delta, and 1.381 μg/mL against Omicron BA.1. However, 1-2C7 had IC50 values of 0.234, 0.270, 0.134, and 0.163 μg/mL against WT, Alpha, Beta, and Delta, respectively, but failed to neutralize BA.1 at the highest concentration used (50 μg/mL) (Fig. 2b). Of note, there appeared to be some differences in IC50 values obtained from the pseudovirus and live virus systems but the overall trend was similar.

## Structural definition of three nanobodies epitopes

We determined crystal structures of 1-2C7, Nb70, and 3-2A2-4 bound to the recombinant RBDs (Fig. 3). 1-2C7 bound to the SARS-CoV-2 Beta RBD (SARS-CoV-2 Beta-RBD) was resolved at 1.8 Å resolution (Fig. 3a and Table S2) whereas the structure of 3-2A2-4 bound to the SARS-CoV-2 wildtype RBD (SARS-CoV-2 WT-RBD) was solved to 2.4 Å resolution (Fig. 3d and Table S2). The crystal structure of Nb70 was determined in two forms. One was the ternary complex of Nb70 and human antibody P2C-1F11 simultaneously bound to the SARS-CoV-2 WT-RBD at 2.4 Å resolution (Fig. 3b and Table S2). The other was the binary complex of Nb70 bound to the SARS-CoV-1 wild type RBD (SARS-CoV-1 WT-RBD) at 2.4 Å resolution (Fig. 3c and Table S2).

The crystal structures showed that both 1-2C7 and Nb70 bound to the cryptic face of RBD and substantially overlapped with that of CR3022 (Fig. 3e–g), consistent with the competition data shown in Fig. 2a. Seven residues in the 1-2C7 epitope (Y369, N370, S375, T376, F377, K378 and P384) and nine in the Nb70 epitope (Y369, F377, K378, C379, Y380, G381, V382, S383 and P384) were also part of the CR3022 epitope[52]. However, as 1-2C7 approached its epitope from the above the receptor-binding motif (RBM) and the epitope deviated upward relative to that of CR3022 (Fig. 3a, e), it was expected to create steric hindrance to ACE2 (Fig. 3e). Y369, S371, F374, F377, K378 and Y508 in the RBD interacted with 1-2C7 through hydrogen bonds and salt bridges (Fig. 3i). Nb70 epitope, on the other hand, was largely confined to the middle of RBD cryptic face and further away from the ACE2 binding site (Fig. 3f). The Nb70 epitopes were highly conserved between SARS-CoV-2 and SARS-CoV-1 (Fig. 3f, g). A total of 17 residues were shared between the two respective epitopes, namely Y369, F374, F377, K378, C379, Y380, G381, V382, S383, R408, A411, P412, G413, Q414, D427, D428, and F429 on the SARS-CoV-2 RBD and Y356, F361, F364, K365, C366, Y367, G368, V369, S370, R395, A398, P399, G400, Q401, D414, D415, and F416 on the SARS-CoV-1 RBD (Fig. 3f and g). These residues interacted with R31, E52, N101, Y103 and D115 of Nb70 through hydrogen bonds, salt-bridges, and van der Waals contacts (Fig. 3k, l, Table S3). For instance, D115 formed hydrogen bonds and salt bridges with K378 and Y380 at the Nb70-SARS-CoV-2 interface and with K365, Y367, R395 and Q401 at the Nb70-SARS-CoV-1 interface (Fig. 3k, l, Table S3). Such conserved epitopes provided a structural basis for the cross-neutralizing activity of Nb70 against SARS-CoV-2 and SARS-CoV-1. By contrast, 3-2A2-4 bound to the bottom part of RBD between the cryptic and the outer face with minimal levels of overlaps with that of the Class 3 antibody S309 (Fig. 3h). The CDR3 residues F102 and F103 penetrated into a hydrophobic pocket and formed

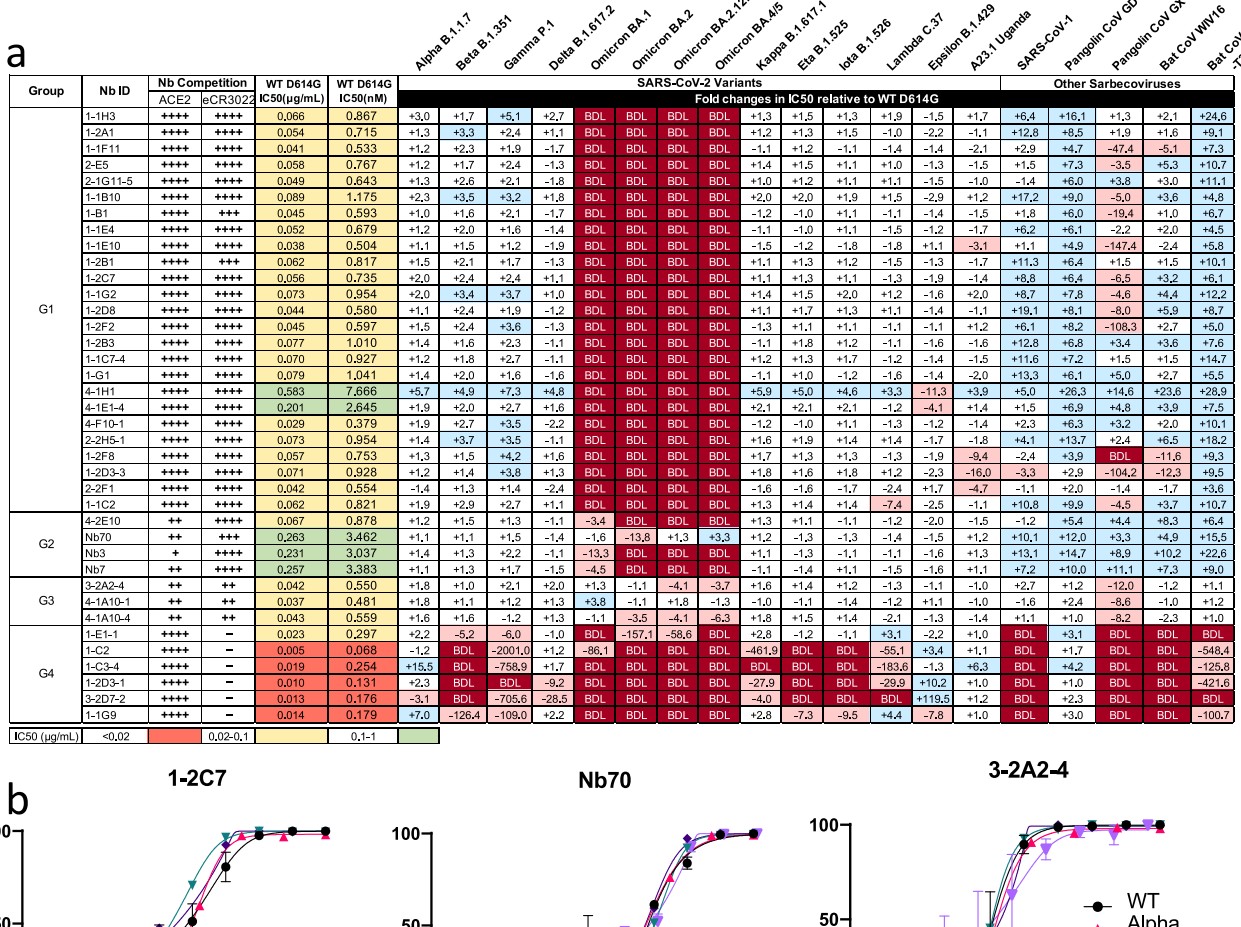

**Fig. 2 | Classification and neutralizing activity of isolated nanobodies against SARS-CoV-2 variants and diverse hACE2-dependent sarbecoviruses.**
**a** Classification of nanobodies into four major groups based on their degrees of competition with ACE2 and control eCR3022 antibody with known epitope specificity, measured by surface plasmon resonance (SPR). "++++" indicates >90% competition, "+++" 60-90%, "++" 30-60%, "+" 10-30%, and "-" no detectable competition. Neutralizing activities of nanobodies in the Fc format against WT D614G were presented by IC50 (µg/mL and nM) while that against SARS-CoV-2 variants and hACE2-dependent sarbecoviruses were in fold-changes relative to that of WT D614G. IC50 values highlighted in salmon indicate <0.02 µg/mL; in yellow 0.02-0.1 µg/mL, and in green >0.1 µg/mL. "-" indicates increased resistance and "+"

indicates increased sensitivity to nanobody neutralization. The fold-changes highlighted in light red indicate that resistance increased at least 3-fold; in light blue, sensitivity increased at least threefold; and in white, resistance or sensitivity increased less than 3-fold. BDL (below detection limit) in dark red indicates that nanobodies at their highest concentration (13.33 µg/mL) failed to reach 50% neutralization. Results were calculated from three independent experiments and each performed in technical duplicates. **b** Neutralizing activity of representative nanobodies from G1, G2, and G3 against authentic SARS-CoV-2 of wildtype (WT) and Alpha, Beta, Delta and Omicron VOCs. Data are presented as mean ± SD. The results shown are representatives of two independent experiments. See also Fig. S2, S3 and Table S1. Source data are provided as a Source Data file.

hydrophobic interactions with RBD residues F338, G339, F342, Y365, V367, L368 and F374 from the core subdomain (Fig. 3j). Upon binding, 3-2A2-4 pushed N343 glycan away from the hydrophobic pocket (Fig. 3j). Such binding pose was distinct from that of S309 and expected to result in different mechanism of action in neutralization. Additional residues such as G339, N343 and D364 but not S371 of SARS-CoV-2 RBD interacted with 3-2A2-4 through hydrogen bonds (Fig. 3j).

Lastly, by analyzing over 8 million SARS-CoV-2 and 46 representative sarbecovirus RBD sequences submitted to GISAID between December 2019 to June 2022, we found that epitope residues of 3-2A2-4 (average 96.64% in SARS-CoV-2 and 91.94% in sarbecoviruses) and Nb70

(average 97.98% in SARS-CoV-2 and 89.44% in sarbecoviruses) were more conserved than that of 1-2C7 (average 92.89% in SARS-CoV-2 and 77.02% in sarbecoviruses) (Fig. 3m, n), providing molecular basis for the broad and potent neutralizing activity of Nb70 and 3-2A2-4 against all viruses tested including Omicron BA.1, BA.2, BA.2.12.1 and BA.4/5.

## 3-2A2-4 nanobody maintains neutralizing activity to Omicron BA.1, BA.2, BA2.12.1 and BA.4/5 subvariants

We next studied the impact of seven single (G339D, S371L, S371F, S373P, S375F, T376A and R408S) and one triple (S371L/S373P/S375F) substitutions found in Omicron subvariants on the neutralizing activity of 1-2C7, Nb70, and 3-2A2-4. These substitutions were located either

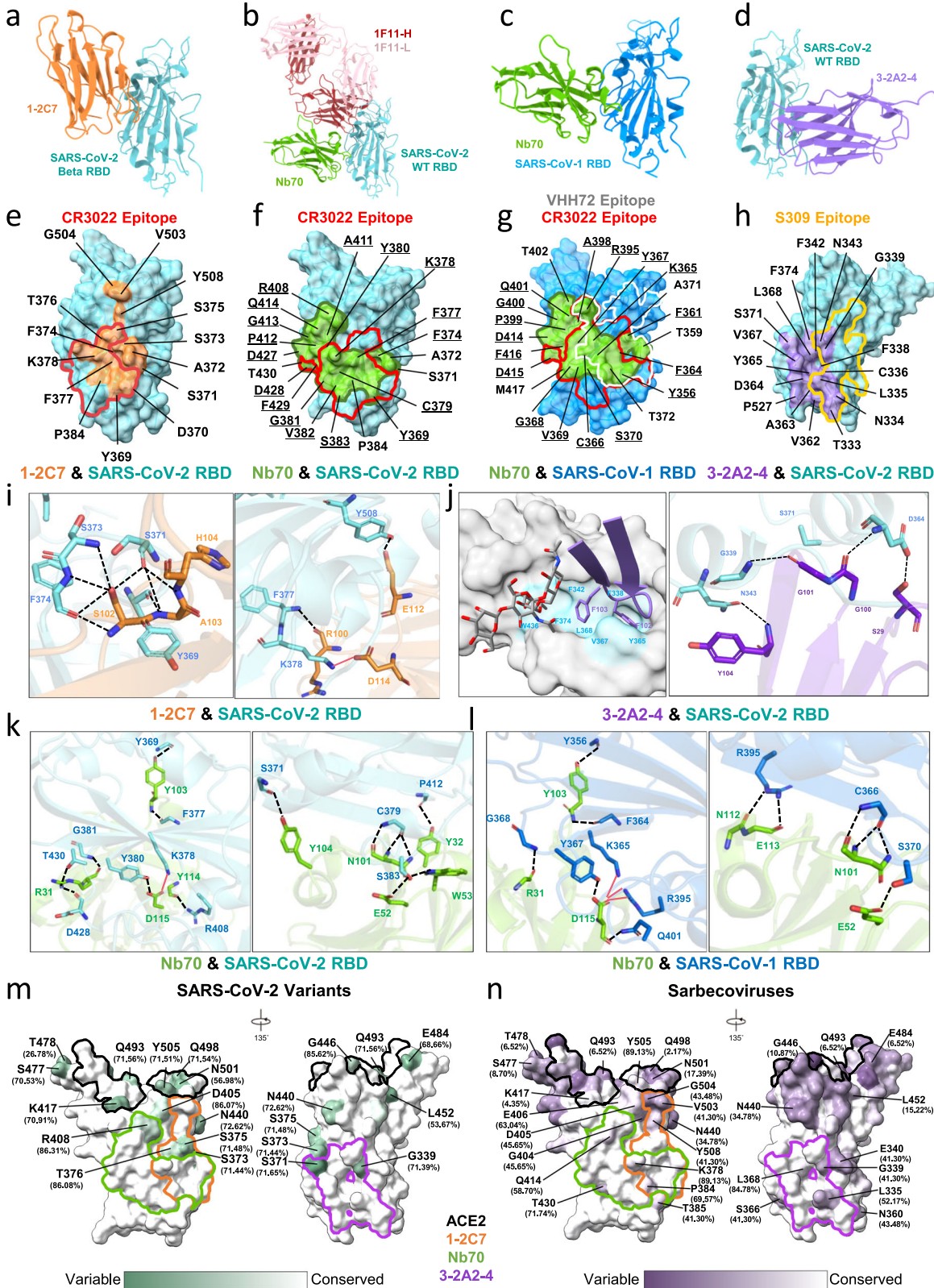

within or proximate to their epitopes (Fig. 4a, d). Pseudoviruses bearing these substitutions were constructed and tested against a serial dilution of 1-2C7, Nb70, and 3-2A2-4 (Fig. 4e, g). As single S375F substitution failed to mediate detectable infection despite multiple attempts, this particular mutant was removed from the subsequent studies. Among the viable mutants tested, 1-2C7 was mostly affected by single S371F substitution, followed by S371L, S373P, and then G339D. The triple

mutations S371L/S373P/S375F resulted in complete loss of activity to the degree that was compatible to BA.1, BA.2, BA.2.12.1, and BA.4/5 (Fig. 4e). Nb70, however, maintained or improved potency to BA.1, BA.2.12.1, and BA.4/5 although substantially reduced to BA.2 (Fig. 4f). Single S371 substitution led to marked reduction (S371L) or completed loss (S371F) of Nb70 activity, although the remaining single (G339D and S373P) or triple (S371L/S373P/S375F) substitutions only had moderate

**Fig. 3 | Crystal structures and epitope of three representative nanobodies bound to SARS-CoV-2 RBD or SARS-CoV-1 RBD. a, d** Crystal structures of 1-2C7 and 3-2A2-4 bound to SARS-CoV-2 Beta and prototype RBD, respectively. **b** Crystal structure of Nb70 and 1F11 bound to SARS-CoV-2 prototype RBD. SARS-CoV-2 RBD is colored in cyan whereas 1-2C7 in orange, Nb70 in green, and 3-2A2-4 in purple. **c** Crystal structure of Nb70 bound to SARS-CoV-1 RBD. Nb70 is shown in green while the SARS-CoV-1 RBD in blue. **e, h** The epitope of 1-2C7 (orange) or of 3-2A2-4 (purple) is respectively depicted on the surface of SARS-CoV-2 RBD. **f** The epitope of Nb70 (green) depicted on the surface of SARS-CoV-2 RBD or **g** on the SARS-CoV-1 RBD. The same residues between SARS-CoV-2 and SARS-CoV-1 epitopes were underlined. The epitope of CR3022 highlighted in red (PDB: 6YM0 on SARS-CoV-2 RBD and PDB: 7JN5 on SARS-CoV-1 RBD) is superposed onto that of **e** 1-2C7 and

**f** Nb70 on SARS-CoV-2 RBD, or onto **g** Nb70 together with that of VHH72 in white on SARS-CoV-1 RBD (PDB: 6WAQ). **h** The epitope of S309 in yellow (PDB: 7R6W) is superimposed onto that of 3-2A2-4. **i** Interactions between 1-2C7 and SARS-CoV-2 Beta RBD, **j** between 3-2A2-4 and SARS-CoV-2 RBD with hydrophobic pocket residues of RBD in light blue., **k** between Nb70 and SARS-CoV-2 RBD, and **l** between Nb70 and SARS-CoV-1 RBD. Levels of residue conservation among **m** over 8 million SARS-CoV-2 RBD sequences and **n** 46 sarbecovirus RBD sequences from GISAID database collected from December 2019 to June 2022. The white surface represents conserved residues, whereas green in **m** or purple in **n** represents variable residues. Several key residues responsible for antibody escape were indicated with levels of conservation. The epitope of ACE2 is outlined in black and the epitope of 1-2C7 is outlined in orange whereas Nb70 in green and 3-2A2-4 in purple.

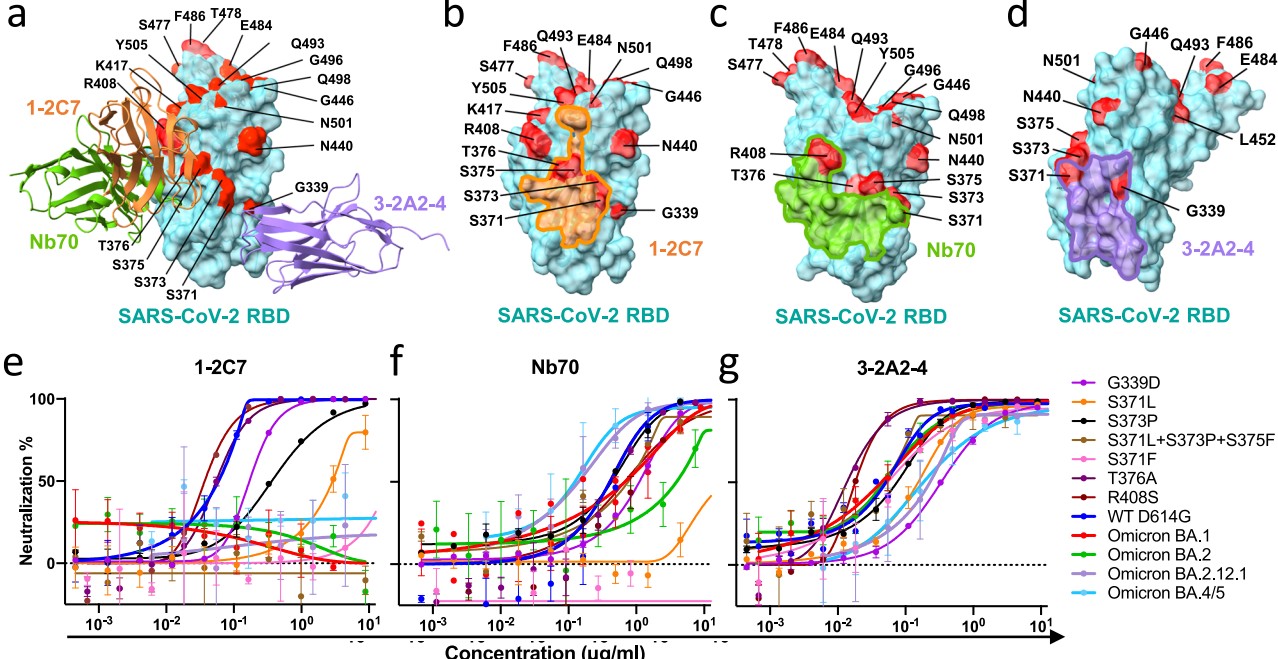

**Fig. 4 | Impact of Omicron mutations on neutralizing activity of the three representative nanobodies. a** Binding modes of 1-2C7 (orange), Nb70 (green), and 3-2A2-4 (purple) to prototype SARS-CoV-2 RBD with the relevant major mutation sites found in Omicron subvariants highlighted in red. **b–d** The footprints of 1-2C7 (orange), Nb70 (green), and 3-2A2-4 (purple) on the surface of prototype

SARS-CoV-2 RBD (cyan), relative to relevant major mutation sites found in Omicron subvariants highlighted in red. **e–g** Neutralizing activity of 1-2C7 Fc, Nb70 Fc, and 3-2A2-4 Fc against pseudoviruses bearing the indicated mutations found in Omicron subvariants. Data are presented as the means ± SD from three independent experiments. Source data **e–g** are provided as a Source Data file.

or no effect at all (Fig. 4f). It was possible that the constellation of S371L/S373P/S375F substitutions somehow restored the epitope structure disrupted by the single S371L substitution, allowing Nb70 to resume binding and exerting neutralizing activity to Omicron variants. By contrast, 3-2A2-4 was the most resilient to Omicron subvariants among the three nanobodies tested (Fig. 4g). 3-2A2-4 remained similar neutralizing activity to BA.1 and BA. 2 compared WT D614G, with IC50 of 0.032 μg/mL, 0.047 μg/mL, and 0.043 μg/mL respectively. Moderately reduction was observed in neutralizing activity to BA.2.12.1 and BA.4/5, with respective IC50 of 0.170 μg/mL and 0.153 μg/mL. Single substitutions such as G339D, S371L, or S373P had only moderate effect while the triple S371L/S373P/S375F substitutions, like that occurred to Nb70, restored neutralizing activity indistinguishable to that of WT D614G (Fig. 4g). Unexpectedly, single substitutions T376A and R408S initially found in BA.2 and BA.4/5 enhanced neutralizing activity of 3-2A2-4, although such effect was not observed against BA.2 and BA.4/5 pseudoviruses.

### Distinct mechanism of action among the three nanobodies
To study the potential mechanism of action of the three nanobodies, we first superimposed the three nanobody structures onto the same

SARS-CoV-2 RBD-ACE2 complex (PDB: 6M0J) (Fig. S4a). As 1-2C7 had steric clash with ACE2 upon binding to the RBD, 1-2C7 likely exerted its neutralizing activity through direct competition with ACE2 for binding to RBD. Nb70 and 3-2A2-4, on the other hand, had no apparent spatial clashes with ACE2, indicating distinct mechanisms from 1-2C7 (Fig. S4a). We then explored the impact of Nb70 and 3-2A2-4 on the formation of a proteinase K-resistant core, a surrogate for transitioning of spike trimer from a pre-hairpin intermediate to a six-helix bundle required for viral-cell fusion[53]. Specifically, the same amount of spike trimer (1 ug) was incubated with Nb70 or 3-2A2-4 in the presence or absence of ACE2 before subjected to trypsin and proteinase K digestion. The formation of proteinase K-resistant core was represented by a 70 kDa band on a western blot gel probed by rabbit anti-S2 polyclonal antibody. As shown in Fig. 5a and b, the presence of ACE2 greatly facilitated the formation of proteinase K-resistant core as opposed to the absence of ACE2 (Lane 4 vs. Lane 3), indicating the critical role of ACE2 in promoting the structural transitioning required for ultimate fusion. Interestingly, in the presence of Nb70, similar enhancement effect was detected, regardless of presence or absence of ACE2 (Lane 5 vs. Lane 6), although both were equivalent to that without Nb70 (in Lane 4 in Fig. 5a, b). These results suggest that binding of Nb70 to the

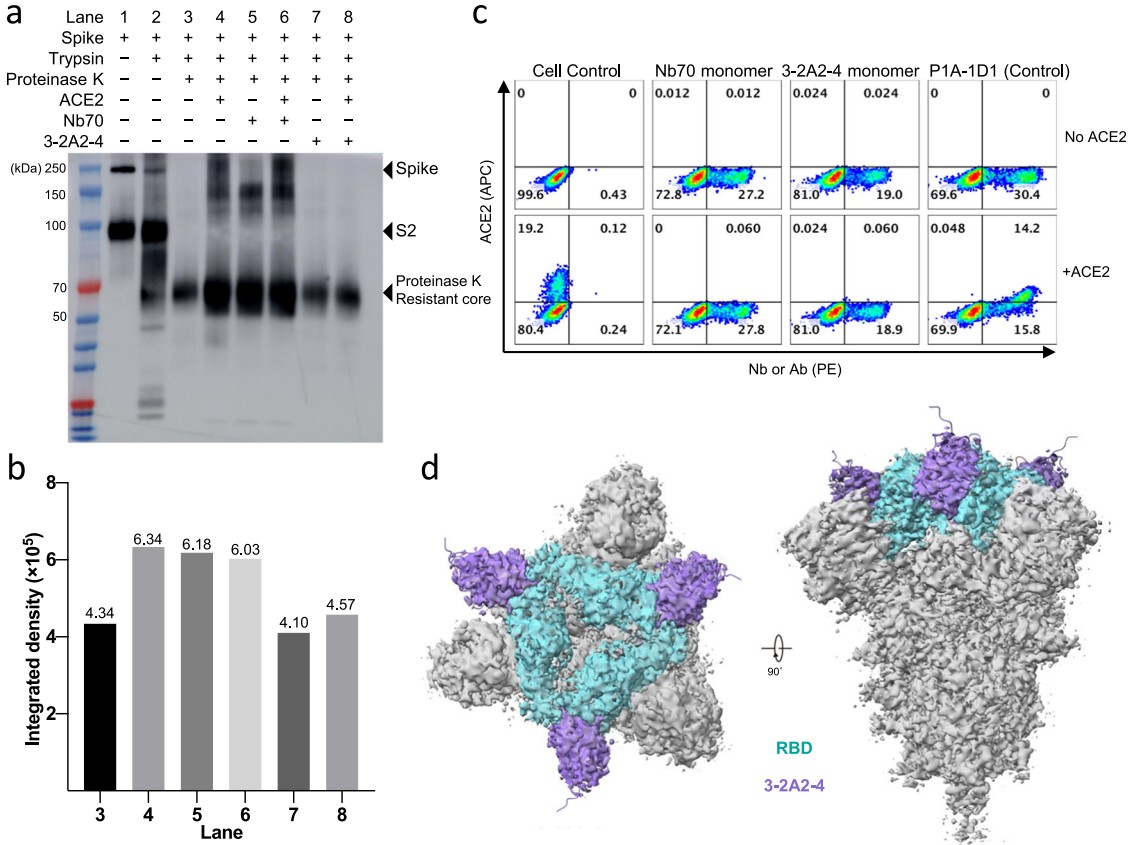

**Fig. 5 | Mechanism of action of Nb70 and 3-2A2-4. a** The effect of monomeric Nb70 and 3-2A2-4 nanobodies on the conformational change of the SARS-CoV-2 S trimer probed by western blotting using an anti-SARS-CoV-2 S2 polyclonal antibody. Refolding to the post-fusion conformation was detected by the appearance of a proteinase K-resistant core. Digestion experiments and western blots were performed in triplicates, and one representative result is shown. **b** The integrated density of proteinase K-resistant core in each lane was calculated by ImageJ. **c** Competitive binding of monomeric Nb70 and 3-2A2-4 nanobodies with soluble ACE2 to prototype SARS-CoV-2 spike measured by cell surface staining. Spike were expressed on the surface on HEK293T, incubated with the Nb70 or 3-2A2-4

nanobody and then ACE2, followed by staining with anti-His tag-PE (nanobodies) and Streptavidin-APC (ACE2) and analyzed by FACS. A human monoclonal antibody P1A-1D1, previously isolated by our group from SARS-CoV-2 convalescent individual and showed minimal competition with ACE2, was used as the antibody control and detected by staining with anti-human IgG Fc-PE. The cell staining was repeated and one representative result is shown. **d** Cryo-EM density maps of the 3-2A2-4 bound to SARS-CoV-2 Omicron BA.1 spike. 3-2A2-4 is shown in purple while the SARS-CoV-2 RBD in cyan and the rest of spike in grey. Source data **a**, **b** are provided as a Source Data file.

cryptic inner face of RBD, like those in the typical class 4 antibodies, might result in allosteric opening of the spike trimer which in turn lead to the formation of six-helix bundle even before reaching the target cells[52,54]. By contrast, in the presence of 3-2A2-4, the formation of proteinase K-resistant core was substantially reduced regardless of ACE2 (Lane 7 vs. Lane 8) to the levels that indistinguishable from the baseline spike (Lane 3). This result indicates that 3-2A2-4, although not directly competing with ACE2 for binding to RBD, might affect the step before and/or after ACE2 binding.

We then measured the competitive binding between Nb70 or 3-2A2-4 and ACE2 to the cell-surface expressed spike trimer of SARS-CoV-2 by FACS (Fig. 5c). Binding of Nb70 and 3-2A2-4 to the spike trimer on the cell surface clearly precluded ACE2 binding, although the control antibody P1A-1D1, previously isolated from a SARS-CoV-2 convalescent individual and shown minimal competition with ACE2 for binding to RBD[55], allowed detectable binding of ACE2 (Fig. 5c). This result indicates that Nb70 and 3-2A2-4 affected ACE2 binding to the spike trimer although Nb70 appeared to trigger premature formation of six-helix bundle while 3-2A2-4 was uncertain. Finally, we used cryo-EM to analyze the structural features of Omicron BA.1 spike trimer bound with 3-2A2-4 to further investigate the potential steps where 3-2A2-4 exerted its inhibition. Out of 947,196 particles collected and applied for initial model and 3D classification, 106,254 particle images

were further refined and processed. Unexpected, all three RBD bound with 3-2A2-4 were in down conformation (Fig. 5d and S5). This was in distinct contrast to that previously reported where the predominantly conformation of Omicron BA.1 spike trimer was with one RBD in the up conformation and the rest two in the down conformation[56,57]. This result indicated that binding of 3-2A2-4 to the spike trimer affected allosteric transitioning of RBD from the down to the up configuration, thereby inhibiting binding to ACE2, the formation of proteinase K-resistant core (see above), and subsequent fusion steps.

As all the three nanobodies contained Fc tail at their C-terminus, we went further to evaluate the impact of Fc tail on levels of competition with soluble ACE2 for binding to RBD (Fig. S4). As expected, 1-2C7, strongly competed with ACE2, regardless with or without Fc tail (Fig. S4 b, e). 3-2A2-4, on the other hand, only competed with ACE2 when containing the Fc tail, suggesting Fc tail accounted for the partial competition with ACE2 (Fig. S4d, g). Most surprisingly, Nb70 demonstrated partial yet convincing levels of competition with ACE2 irrespective of Fc tail (Fig. S4c, f). Similar result on CR3022 competition with ACE2 has also been reported[58]. These results, therefore, indicate that Nb70, CR3022, or perhaps some class IV antibodies bind to the inner face of RBD to induce premature-S1 shedding, which indirectly affect ACE2 binding to RBD. This would explain why Nb70, with or without Fc, appeared to have a similar competition profile with ACE2.

In summary, we have demonstrated distinct mechanisms of three nanobodies (1-2C7, Nb70, and 3-2A2-4) in competing with ACE2 thereby in neutralizing SARS-CoV-2. 1-2C7 acts through direct competition for binding to RBD. Nb70 functions indirectly through inducing premature-S1 shedding and triggering pre-mature formation of proteinase K-resistant core before viral interaction with ACE2 on the target cells. 3-2A2-4, on the other hand, appears to suppress RBD in the "down" conformation thereby blocking RBD transitioning to the "up" conformation required for ACE2 binding and viral entry.

### 3-2A2-4 protects K18-hACE2 mice from infection with authentic SARS-CoV-2 Omicron and Delta

We next studied the protective potential of 3-2A2-4 against infection of authentic Omicron and Delta variants in a K18-hACE2 mouse model of SARS-CoV-2 infection, as previously described (Fig. 6a)[59]. Specifically, the mice were intraperitoneally administered with 3-2A2-4 at a dose of 10 mg/kg body weight 24 hours prior to intranasal challenge with $1.7\times10^3$ plaque-forming units (PFU) of authentic SARS-CoV-2 Omicron or $10^3$ PFU of SARS-CoV-2 Delta. The animals were then monitored daily throughout the following 14 days for their body weight and survival. Half of the animals were euthanized on day 3 post-challenge to obtain lung tissues for viral load and histopathological analysis.

In SARS-CoV-2 Omicron challenged groups, one out of the six untreated animals succumbed to the disease on day 11 post infection whereas all 3-2A2-4 treated mice remained healthy and survived infection (Fig. 6b). The changes in body weight followed the similar trend of survival, with moderate loss in untreated compared to relative stability in 3-2A2-4 treated animals, although animals in both groups experienced minor body weight loss during the first 6 days after challenge (Fig. 6c). No detectable levels of live viruses were found in the lungs of 3-2A2-4-treated mice on day 3 post challenge while that in untreated mice reached an average 796.7 PFU/tissue (Fig. 6d). Immunohistochemistry analysis revealed that the lung tissue of 3-2A2-4-treated mice remained intact and no viral antigen-positive cells were detected (Fig. 6h). By contrast, the lung sections of untreated mice presented moderate damage and inflammation with marked infiltration of inflammatory cells. Infected cells were readily detectable using anti-N protein-specific antibody (Fig. 6h).

In SARS-CoV-2 Delta-challenged groups, untreated animals exhibited faster disease progression and greater severity compared to those in the Omicron challenged group, indicating Delta was more pathogenic than Omicron in this model of SARS-CoV-2 infection, in complete agreement with those recently reported[13,14]. All untreated animals succumbed to diseases by day 6 after the challenge and associated with severe body weight loss (Fig. 6e, f). By contrast, 3-2A2-4-treated group remained fully protected and maintained stable body weights, except for one mouse had to be euthanized on day 6 after infection due to the requirement of experimental protocol when body weight fell below 75% of baseline (Fig. 6e, f). No detectable levels of viruses in the lungs were found in 3-2A2-4-treated mice while that in untreated mice reached as high as $8.1\times10^4$ PFU/tissue on average (Fig. 6g). The lung sections from untreated animals showed severe lung damage and inflammation with marked infiltration of inflammatory cells. A large number of viral antigen-positive pneumocytes were clearly visible (Fig. 6i). By contrast, in 3-2A2-4-treated mice, lung tissue remained relatively intact and well-defined. Collectively, these results indicated that the broad and potent neutralizing nanobody 3-2A2-4 conferred strong protection in vivo against challenge of authentic SARS-CoV-2 Omicron and Delta.

### Discussion

The rapid turnover of antigenically distinct SARS-CoV-2 VOCs particularly Omicron subvariants have resulted in the substantial reduction and loss of activity of many therapeutic antibodies and vaccines[7–9,11]. As unusually large number of mutations are present in the spike of these Omicron subvariants, one of the urgent questions needs to be addressed is whether conserved and protective epitopes still exist on the spike trimer that can be targeted for the development of broad and potent antibody therapies and vaccines. Here, we report on the isolation and characterization of a unique group of nanobodies from immunized alpaca, exemplified by 3-2A2-4, that have potency against Omicron subvariants BA.1, BA.2, BA.2.12.1, and BA.4/5, SARS-CoV-1, and the key representative coronaviruses from bats and pangolins. The IC50 reached as low as 0.042 µg/mL or 0.550 nM against WT D614G and remained relatively stable across the entire panel of 19 viruses tested. Although neutralization assays vary from different laboratories, the levels of neutralization suggest 3-2A2-4 is amongst the broadest and most potent nanobodies described to date[33–46]. Such impressive in vitro activity was translated into strong protectivity against infection of authentic Delta and Omicron in the K18-hACE2 mice model. Crystal structure analysis revealed a novel and highly conserved epitope of 3-2A2-4 located between the cryptic and outer face of RBD, distinctive from the ACE2 binding site. Mechanistically, 3-2A2-4 appeared to suppress all three RBD in the "down" conformation, interfering their transitioning to the "up" conformation required for binding to receptor ACE2. This is consistent with biochemical analysis where 3-2A2-4 was found to substantially reduced the formation of proteinase K-resistant core. Of note, the ability of 3-2A2-4 to suppress RBD in the "down" configuration was not due to the 6P and GSAS substitutions used to stabilize the overall structure of the spike trimer, as a previous cryo-EM study of the Omicron S trimer containing GSAS mutations at the furin cleavage site and 6P mutations showed only open conformation with one up-RBD after 3D classification (PMID: 35120603). However, when bound with 3-2A2-4 in our study, all subsets of spikes after 3D classification were in the closed state (Fig. S5)[56,60,61]. Collectively, the unique binding pose and mechanism of action provide a structure and function basis for 3-2A2-4 in withstanding the major mutant residues that are commonly found in major VOCs and compromised many therapeutic antibodies approved for EUA[7,11]. The nanobody and the epitope identified here should provide an important reference for the development of next-generation antibody therapies and vaccines against diverse sarbecoviruses including Omicron BA.4/5 that responsible for the current wave of infections.

Owing to their nanoscale and extended CDR3, nanobodies are expected to penetrate deeper into the spike trimer and access to hidden epitopes that are less frequently exposed and/or unreachable by conventional antibodies. This is particularly true for the cross-neutralizing SARS-CoV-2 and SARS-CoV-1 nanobodies characterized here. Virtually all nanobodies in the G1 and G2 were able to compete with a published nanobody eCR3022 known to recognize the hidden epitope accessible only when RBD is in the "up" conformation[35]. By contrast, the nanobodies in the G3 bound to the epitopes that are readily accessible regardless of up or down conformations of RBD. Interestingly, by screening and characterizing hundreds of monoclonal antibodies from convalescent or vaccinated individuals, a small but convincing number of cross-neutralizing antibodies with similar epitope specificity have also been found[61–71]. The epitopes recognized by the G1, G2, and G3 nanobodies are therefore not only the viable targets in alpaca but also in humans. In particular, the G1 and G2 nanobodies would fall into the inner face antibodies exemplified by CR3022, H014, S2X259, COVA1-16, and CV2-75 whereas the G3 nanobodies with the escarpment face antibody 47D11 identified in mice (Fig. S6)[54]. Additional cross-neutralizing SARS-CoV-2 and SARS-CoV-1 antibodies targeting to other faces of RBD have also been identified such as those to the cliff face (S2H97 and 6D60), top face (S2X146), and outer face (S309 and BG10-19) (Fig. S6). Recently, a human antibody with broad reactivity to human beta-coronaviruses has been isolated that targets the conserved S2 stem-helix, raising the possibility of development of pan-β-CoV therapies and vaccines[72–74]. More importantly, these cross-neutralizing and pan-β-CoV antibodies can be

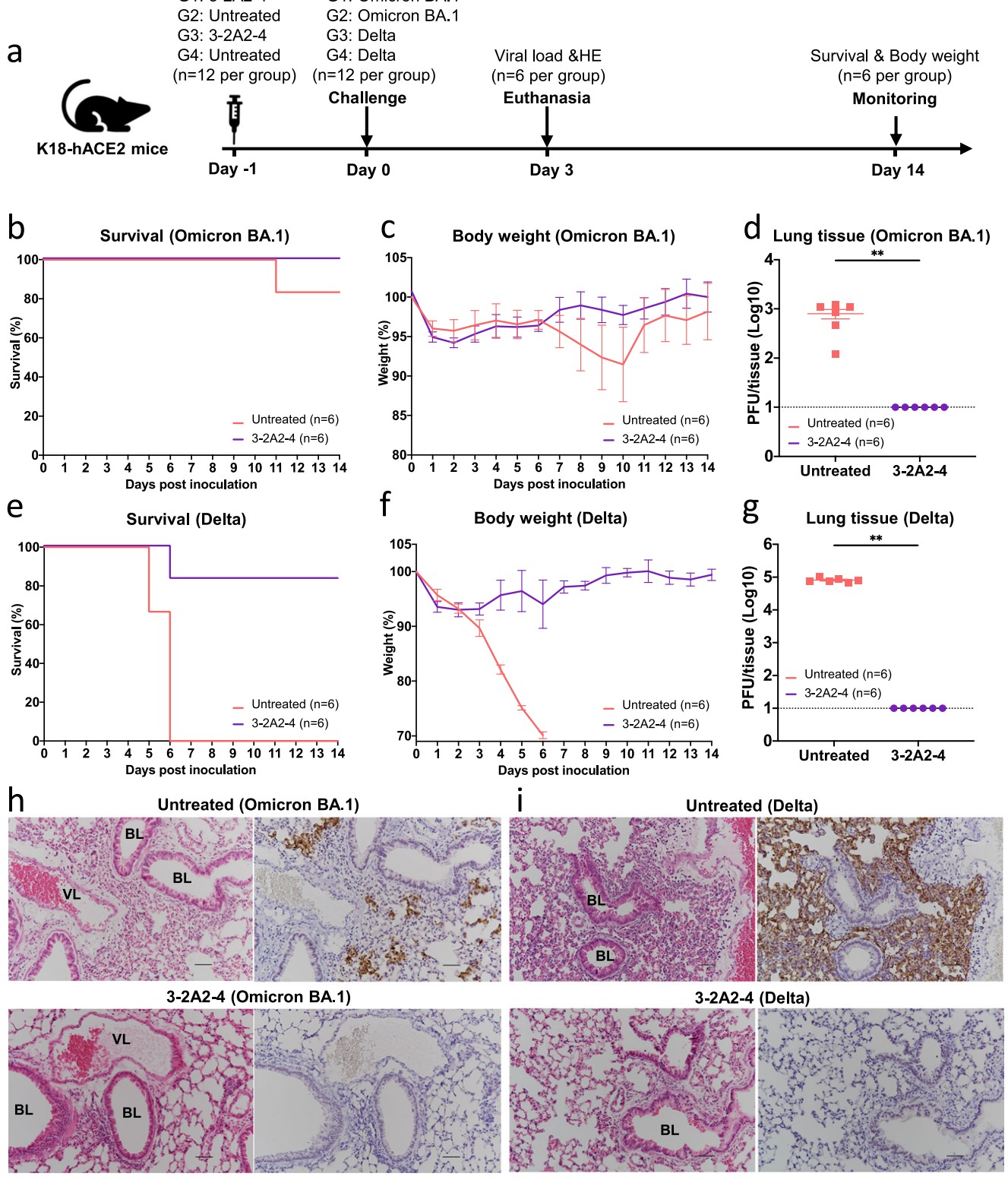

**Fig. 6 | Efficacy of 3-2A2-4 prophylaxis against infection of the authentic SARS-CoV-2 Omicron and Delta in K18-hACE-2 mice. a** Experimental schedule for nanobody prophylaxis. Eight-week-old K18-hACE2 transgenic female mice were intraperitoneally administered with 10 mg/kg body weight of 3-2A2-4 in the format of Fc or untreated 1 day prior to challenge with $1.7 \times 10^3$ plaque-forming units (PFU) infectious SARS-CoV-2 Omicron or $10^3$ PFU Delta via the intranasal route. The survival percentage **b** and **e** and body weight **c** and **f** were recorded daily after infection until the occurrence of death or until the end of experiment. The viral load in the lung tissue **d** and **g** was tested by plaque assays in the tissue homogenates at 3 days post inoculation. The limit of detection of plaque assay is 10 PFU/tissue. Data are presented as the means ± SEM. The differences between 3-2A2-4 group and untreated group are statistically significant with two-tailed *p* value = 0.0022 (***P* < 0.01, Mann–Whitney test) in both Omicron BA.1 and Delta challenge. **h**, **i** HE and IHC staining of lung tissue from 3-2A2-4-treated or untreated mice at 3 days post inoculation. VL vascular lumen, BL bronchiolar lumen. Scale bars, 50 µm. Results presented are representatives of each group (*n* = 6). Source data **b**–**g** are provided as a Source Data file.

substantially boosted by mRNA vaccines, particularly in individuals with pre-existing SARS-CoV-1 or SARS-CoV-2 infection[49,75–77]. Comprehensive characterization of these antibodies will provide deeper insights into their ontogeny and potential ways of inducing broader and more effective protection against the circulating and future variants.

Finally, our study has some inherent shortcomings. Given the nature of immune response in alpaca is likely different from that in human, the antibody responses and the epitopes recognized by the nanobodies may not be exactly reflective of that in humans. Furthermore, our nanobody protection experiments were performed exclusively in K18-hACE2 mice, which may not represent that against SARS-CoV-2 infection in humans. Of note, as Omicron and Delta resulted in rather distinct pattern of diseases and viral replication in the animal model, the protection efficacies against the two variants should therefore not be compared. Future studies in NHP and humans would be highly desirable to verify and validate the protection results.

## Methods

### Cell lines

HEK293T cells (ATCC, CRL-3216) and HeLa cells expressing hACE2 were kindly provided by Dr. Qiang Ding at Tsinghua University. Both of these cell lines were maintained at 37 °C in 5% CO2 in Dulbecco's minimal essential medium (DMEM) containing 10% (v/v) heat-inactivated fetal bovine serum (FBS) and 100 U/mL penicillin–streptomycin. FreeStyle 293 F cells (Thermo Fisher Scientific, R79007) were maintained at 37 °C in 5% CO2 in SMM 293-TII expression medium (Sino Biological, M293TII). Sf9 cells (ATCC) were maintained at 27 °C in Sf-900 II SFM medium. Hi5 cells (ATCC) were maintained at 27 °C in SIM HF medium.

### Expression and purification of recombinant proteins

The genes encoding the ectodomain of S trimer and S2 trimer of prototype SARS-CoV-2 Wuhan-Hu-1 strain (GenBank: MN908947.3) were constructed as previously reported[10]. Both S trimer (residues M1-Q1208) and S2 trimer (residues S686-Q1208) contained proline substitutes at residue 986 and 987, a foldon trimerization motif, and a strep tag at C-terminal. Additional 'GSAS' substitution at furin cleavage site (residues 682-685) was also introduced into the S trimer construct to improve overall stability. Both S trimer and S2 trimer were expressed in the FreeStyle 293 F cells and purified by Strep-Tactin Sepharose (IBA Lifesciences 2-1201-025) followed by gel filtration chromatography (GE Healthcare). The gene encoding the ectodomain of SARS-CoV-2 Omicron BA.1 spike (GenBank ID: ULC25168.1) was constructed and contained the following modifications, proline substitutes at residue 817, 892, 899, 942, 986 and 987, "GSAS" substitutions at furin site, a foldon tag for trimerization, and a strep tag at C-terminal for purification. The trimeric Omicron BA.1 spike was expressed in the FreeStyle 293 F cells and purified by Strep-Tactin Sepharose followed by gel filtration chromatography. The S trimer of SARS-CoV-1, RBD of SARS-CoV-1, SARS-CoV-2 prototype and Beta strain, and the N-terminal peptidase domain of human ACE2 were expressed using the Bac-to-Bac Baculovirus Expression System (ThermoFisher 10359016) and Ni-NTA resin (QIAGEN 30230) followed by gel filtration chromatography, as previously reported[78]. Briefly, the RBD of SARS-CoV-2 prototype (residues Arg319−Lys529) with an N-terminal gp67 signal peptide for secretion and a C-terminal 6×His tag for purification was expressed using Hi5 cells and purified by Ni-NTA resin followed by gel filtration chromatography in HBS buffer (10 mM HEPES, pH 7.2, 150 mM NaCl). The RBD of SARS-CoV-1 (residues Arg306−Leu515) and N-terminal peptidase domain of human ACE2 (residues S19−D615) were expressed and purified by the same protocol as used for the RBD of SARS-CoV-2 prototype. The SARS-CoV-2 NTD protein was purchased from SinoBiological (40591-V49H).

### Immunization of alpaca, construction of yeast display VHH library, and isolation of VHH yeasts specific for SARS-CoV-2 and SARS-CoV-1 spikes

The animal experiment protocol involving immunization, collection of blood samples, and construction of VHH library was approved by IACUC at NBbiolab, Inc. in Chengdu, China. The immunization procedure involved three-time subcutaneous injections of 200 µg recombinant RBD of prototype SARS-CoV-2 in Freund adjuvant, one-time subcutaneous injection of 10[11] viral particle AdC68-19S vaccine expressing the prototype S trimer[48], and two-time subcutaneous injections of 200 µg recombinant S-2P protein of prototype SARS-CoV-2 in Freund adjuvant. Seven days after the last immunization, blood samples were collected to isolate peripheral blood lymphocytes and plasma. Total RNA extracted from the peripheral blood lymphocytes was used as the template for the synthesis of the first strand cDNA, using oligo dT as a primer. VHH sequences were amplified by PCR, cloned into a yeast surface display vector pYD1, and transfected into the electrocompetent EBY100 cells. The yeast library was first grown in SDCAA media at 30 °C for 48 h. At the exponential growth phase, the yeast library was transferred to SGCAA media for induction of VHH expression at 20 °C for 36 h. The yeast clones displaying VHHs specific to SARS-CoV-2 or SARS-CoV-1 spikes were enriched by one round of MACS biopanning followed by an additional round of FACS biopanning. Specifically, the induced yeast library was collected and incubated with 100 nM SARS-CoV-2 S-2P protein or SARS-CoV-1 S protein on ice for 30 min. The yeast library was washed with cold PBS containing 1%FBS for three times and incubated with streptavidin microbeads on ice for 10 min. The mixture was passed through LS column and the S protein-positive yeasts were harvested for further culture and induction of VHH expression. Once again, induced yeasts were incubated with 100 nM SARS-CoV-2 S-2P protein or SARS-CoV-1 S protein on ice for 30 min. After extensive wash with cold PBS + 1%FBS, the yeast clones were incubated with HA-Tag (6E2) mouse monoclonal antibody conjugated with Alexa Fluor® 488 (Cell Signaling 2350 S, 1:100 dilution) and streptavidin conjugated with PE Conjugate (eBioscience 12-4317-87, 1:200 dilution) on ice for 30 min. The yeast clones were washed three times with cold PBS containing 1%FBS before analyzed by FACS using Aria II (BD Biosciences). Positive yeast clones were sorted and used for nanobody cloning.

### Molecular cloning and expression of VHH

The sequences encoding various VHHs were amplified from the sorted yeast clones and inserted into two different expression vectors depending on the subsequent study objectives. One was to express VHH in conjunction with human IgG1 Fc fragment to evaluate binding and neutralizing activity of VHHs. The other was to express VHH with a 6×His tag for crystal structural analysis. For the former, VHH genes were cloned into the multiple cloning sites of pMD18T containing the upstream CMV promoter, the secretory signal sequence from the mouse Ig heavy chain, and the downstream human IgG1 Fc gene fragment and SV40 poly (A) signal sequence. For the latter, selected VHH genes were cloned into pVRC8400 vector with a 6×His tag. Expression and production of nanobodies were conducted by transfecting the expression vectors into the FreeStyle 293F cells using polyethyleneimine (PEI) (Polysciences 23966). After approximately 96 h, nanobodies containing the human IgG1 Fc in the culture supernatant were captured by AmMag Protein A Magnetic Beads (Genscript L00695) and eluted by 0.1 M Glycine pH 3.0. Nanobodies with 6×His tag were captured by Ni-NTA Agarose (QIAGEN 30230) and eluted by 500 nM Imidazole. All nanobodies were purified by gel-filtration chromatography with Superdex 200 High-Performance column (GE Healthcare). The exact concentration was determined by nanodrop 2000 Spectrophotometer (Thermo Scientific).

## Production of SARS-CoV-2 and sarbecovirus pseudoviruses

Pseudoviruses carrying the full-length spike envelope of prototype SARS-CoV-2 and VOCs, and sarbecoviruses were generated as previously reported[10]. Specifically, human immunodeficiency virus backbones expressing firefly luciferase (pNL4-3-R-E-luciferase) and pcDNA3.1 vector encoding either SARS-CoV-2 or sarbecovirus spike proteins were co-transfected into the HEK293T cells (ATCC). Forty-eight hours later, pseudoviruses in the viral supernatant were collected, centrifuged to remove cell lysis, and stored at -80 °C until use. The wildtype pseudovirus used throughout the analysis was the prototype strain (GenBank: MN908947.3) with a D614G mutation (WT D614G). The Alpha variant (Pango lineage B.1.1.7, GISAID: EPI_ISL_601443) included a total of 9 reported mutations in the spike protein (69-70del, 144del, N501Y, A570D, D614G, P681H, T716I, S982A and D1118H). The Beta variant (Pango lineage B.1.351, GISAID: EPI_ISL_700450) included 10 identified mutations in the spike such as L18F, D80A, D215G, 242-244del, S305T, K417N, E484K, N501Y, D614G and A701V. The Gamma variant (Pango lineage P.1, GISAID: EPI_ISL_792681) had 12 reported mutations in the spike including L18F, T20N, P26S, D138Y, R190S, K417T, E484K, N501Y, D614G, H655Y, T1027I and V1176F. The Delta variant (Pango lineage B.1.617.2, GISAID: EPI_ISL_1534938) included 10 reported mutations in the spike such as T19R, G142D, 156-157del, R158G, A222V, L452R, T478K, D614G, P681R, D950N. The Omicron BA.1 variant (Pango lineage BA.1, GISAID: EPI_ISL_6752027) was constructed with 34 mutations in the spike such as A67V, 69-70del, T95I, G142D, 143-145del, 211del, L212I, ins214EPE, G339D, S371L, S373P, S375F, K417N, N440K, G446S, S477N, T478K, E484A, Q493R, G496S, Q498R, N501Y, Y505H, T547K, D614G, H655Y, N679K, P681H, N764K, D796Y, N856K, Q954H, N969K, and L981F. The Omicron BA.2 variant (Pango lineage BA.2, GISAID: EPI_ISL_8515362) was constructed with 29 mutations in the spike such as T19I, 24-26del, A27S, G142D, V213G, G339D, S371F, S373P, S375F, T376A, D405N, R408S, K417N, N440K, S477N, T478K, E484A, Q493R, Q498R, N501Y, Y505H, D614G, H655Y, N679K, P681H, N764K, D796Y, N969K and Q954H. The Omicron BA.2.12.1 variant (Pango lineage BA.2.12.1, GISAID: EPI_ISL_12560123) was constructed with 30 mutations in the spike such as T19I, 24-26del, A27S, G142D, V213G, G339D, S371F, S373P, S375F, T376A, D405N, R408S, K417N, N440K, L452Q, S477N, T478K, E484A, Q493R, Q498R, N501Y, Y505H, D614G, H655Y, N679K, P681H, N764K, D796Y, N969K and Q954H. The Omicron BA.4/5 variant (Pango lineage BA.4/5, GISAID: EPI_ISL_12559461) was constructed with 30 mutations in the spike such as T19I, 24-26del, A27S, G142D, V213G, G339D, S371F, S373P, S375F, T376A, D405N, R408S, K417N, N440K, L452Q, S477N, T478K, E484A, F486V, Q498R, N501Y, Y505H, D614G, H655Y, N679K, P681H, N764K, D796Y, N969K and Q954H. The Kappa variant (Pango lineage B.1.617.1, GISAID: EPI_ISL_1384866) was constructed with 8 mutations in the spike such as T95I, G142D, E154L, L452R, E484Q, D614G, P681R and N1071H. The Eta variant (Pango lineage B.1.525, GISAID: EPI_ISL_2885901) included 8 mutations in the spike such as Q52R, A67V, 69-70del, Y144del, E484K, D614G, Q677H and F888L. The Iota variant (Pango lineage B.1.526, GISAID: EPI_ISL_2922249) was constructed with 6 mutations in the spike such as L5F, T95I, D253G, E484K, D614G and A701V. The Lambda variant (Pango lineage C.37, GISAID: EPI_ISL_2930541) was constructed with 8 mutations in the spike such as G75V, T76I, R246N, 247-253del, L452Q, F490S, D614G and T859N. The Epsilon variant (Pango lineage B.1.429, GISAID: EPI_ISL_2922315) was constructed with 4 mutations in the spike such as S13I, W152C, L452R and D614G. The variant with Pango lineage A23.1, GISAID: EPI_ISL_2690464) was constructed with 4 mutations in the spike such as F157L, V367F, Q613H and P681R. The G339D, S371L, S371F, S373P, T376A, R408S and S371L + D373P + S375F were constructed with a D614G mutation based on the WT strain. For sarbecoviruses, the cDNAs encoding the SARS-CoV-1 S glycoprotein (NCBI Accession NP_828851.1), Pangolin CoV GD spike (GenBank: QLR06867.1), Pangolin CoV GX spike (GenBank: QIA48614.1), Bat

SARS-like coronavirus WIV16 (GenBank: ALK02457.1), Bat SARS-like RaTG13 spike (GenBank: QHR63300.2), Bat SARS-like ZC45 (GenBank: AVP78031.1), Bat SARS-like ZXC21 (GenBank: AVP78042.1) and Bat SARS-like coronavirus RsSHC014 (GenBank: AGZ48806.1) were synthesized with codons optimized for protein expression (Genwiz Inc., China) and verified by sequencing.

## Neutralization activity of nanobodies against pseudoviruses and live SARS-CoV-2

Neutralizing activity of nanobodies were determined using SARS-CoV-2 pseudovirus and authentic live virus as previously reported[55]. Briefly, nanobodies were 3-fold serially diluted in 96-well cell culture plates, mixed with SARS-CoV-2 pseudovirus, and incubated at 37 °C for 1 h. HeLa-ACE2 cells were then added to the mixture of nanobody-pseudovirus, incubated at 37 °C for an additional 48 h, and lysed for measuring luciferase-activity. The IC50 values were calculated based on the reduction of 50% relative light units (Bright-Glo Luciferase Assay Vector System, Promega, E2650) compared to the virus-only control, using Prism 8.0 (GraphPad Software Inc., USA).

For the authentic live virus assay, we used the focus reduction neutralization test (FRNT) performed in a certified BSL3 facility at Shenzhen Third People's Hospital, China. Briefly, serial dilutions of nanobodies were mixed with various authentic SARS-CoV-2 and incubated for 1 h at 37 °C. The mixtures were then transferred to 96-well plates seeded with Vero E6 cells and incubated for 1 h at 37 °C. After changing the medium, the plates were incubated at 37 °C for an additional 24 h. The cells were then fixed, permeabilized, and incubated with cross-reactive rabbit anti-SARS-CoV-N IgG (SinoBiological 40588-T62, 1:1000 dilution) for 1 h at room temperature before adding Horseradish Peroxidase (HRP)-conjugated goat anti-rabbit IgG (Heavy chain + Light chain) antibody (TransGen Biotech HS101-01, 1:2000 dilution). The reactions were developed using KPL TrueBlue peroxidase substrate (Seracare Life Sciences 5510-0030). The number of SARS-CoV-2 foci was quantified using an EliSpot reader (Cellular Technology Ltd. USA). The authentic SARS-CoV-2 used in the assay included the WT, Alpha, Beta, Delta, and Omicron BA.1 and were isolated locally. Their whole genome sequences have been deposited into China National Center for Bioinformation, with accession numbers GWHXXXX00000000 for WT strain, GWHBFWX01000000 for Alpha strain, GWHBDSE01000000 for Beta strain, GWHBFWZ01000000 for Delta strain, and GWHBIRY01000000 for Omicron BA,1 strain, which are publicly accessible at https://ngdc.cncb.ac.cn/gwh. The viruses of Beta strain and Delta strain were gifted from Guangdong Provincial Center for Disease Control and Prevention, Guangdong Center for Human Pathogen Culture Collection (GDPCC).

## Phylogenetic tree and genetic analysis of nanobodies

Genetic relatedness and clustering among isolated nanobodies were analyzed by neighbor-joining phylogenetic trees were generated using MEGA version 10.1.8 with 1000 bootstrap replicates. The IMGT/V-QUEST program (http://www.imgt.org/IMGT_vquest/vquest) was used to analyze the germline gene and the loop lengths of complementarity determining region 3 (CDR3) of each nanobody. Chord diagrams showing the germline gene usages and V/J gene pairing were analyzed and presented by the R package circlize version 0.4.13[79]. The width of the linking arc is proportional to the number of nanobodies identified. Sequence logo were plotted using Python package Logomaker[80].

## Nanobody competition with ACE2 and control antibody measured by SPR

The competitive binding to SARS-CoV-2 RBD between nanobodies and ACE2 and control antibody eCR3022.7 were analyzed using SPR (Biacore 8 K, GE Healthcare). Specifically, the prototype SARS-CoV-2 RBD

was immobilized to a CM5 sensor chip (Cytiva BR100530) via the amine group for a final RU around 250. In the first round, nanobodies (1 μM) or eCR3022 (1 μM) were injected onto the chip for 120 s to reach the steady state for binding. In the second round, nanobodies (1 μM) or eCR3022 (1 μM) were injected onto the chip for 120 s followed by the injection of ACE2 (2 μM) or nanobodies (1 μM) for additional 120 s. In the third round, running buffer was injected for 120 s followed by injection of ACE2 (2 μM) or nanobodies (1 μM) for another 120 s. The sensorgrams of the three rounds were aligned from 120 to 240 s in Biacore 8 K Evaluation software (GE Healthcare). The blocking efficacy was determined by a comparison of the response units with and without prior antibody injection.

### Western Blot analysis for proteinase-K resistance core

SARS-CoV-2 spike (residues M1-Q1208), containing a foldon trimerization motif and a strep tag at C-terminal were expressed in the FreeStyle 293 F cells and purified by Strep-Tactin Sepharose followed by gel filtration chromatography. 0.4 μg of the spike was incubated with 2 μg of the ACE2 or 2 μg of monomeric nanobodies for 1 h on ice. Trypsin (final concentration of 5 μg/mL, ThermoFisher 27250018) was then added to this mixture and incubated for an additional 20 min at 37 °C. Subsequently, the samples were subjected to proteinase-K (ThermoFisher AM2546) digestion with final concentration of 100 μg/mL and incubated 20 min at 37 °C. 4×SDS-PAGE loading buffer was then added to all samples prior to boiling at 100 °C. Samples were run on a 4–12% gradient Tris-MOPS-Gel (GenScript M00653) and transferred to polyvinylidene fluoride membranes. An anti-SARS-CoV-2 S2 polyclonal antibody (SinoBiological T40590-T62, 1:2000 dilution) and an HRP-conjugated anti-rabbit secondary antibody (Promega W4011, 1:4000 dilution) were used for Western blotting. The image was developed by AI600 and the intensity of proteinase-K resistant core was estimated by ImageJ 1.53k.

### Cell surface staining

HEK 293 T cells were transfected with expression plasmids encoding SARS-CoV-2 prototype spike, and incubated at 37 °C for 24 h. Cells were digested from the plate with trypsin and distributed onto 96-well plates. Cells were washed twice with 200 μL staining buffer (PBS with 1% heated-inactivated fetal bovine serum (FBS)) between each of the following steps. First, cells were stained with each monomeric nanobody (20 μg/mL) at 4 °C for 20 min and then ACE2 (20 μg/mL) at 4 °C for 20 min. Anti-His tag-PE (Miltenyi Biotec 130-120-787, 1:200 dilution) and Streptavidin-APC (eBioscience 17-4317-82, 1:200 dilution) were added and incubated at 4 °C for 20 min. After extensive washes, the cells were resuspended and analyzed by BD LSRFortassa (BD Biosciences, USA) and FlowJo 10 software (FlowJo, USA). A human monoclonal antibody P1A-1D1, previously isolated by our group from SARS-CoV-2 convalescent individual and showed minimal competition with ACE2[55], was used as the antibody control and detected by staining with anti-human IgG Fc-PE (Biolegend 410708, 1:50 dilution). HEK 293 T cells with mock transfection were stained as background control. The number of positive cells in the selected gates were calculated (Fig. S4h).

### Crystallization and data collection

To obtain the complex of RBD bound to nanobodies, RBD was incubated with each nanobody for 1 h on ice in HBS buffer. The mixture was then purified using gel filtration chromatography. Fractions containing the complex were pooled and concentrated to 10 mg/mL. Crystals were successfully grown at room temperature in sitting drops, over wells containing 0.2 M Ammonium sulfate, 0.1 M Bis-Tris pH 4.4, 21% w/v Polyethylene glycol 3,350 for Nb70-SARS-CoV-1 RBD, 0.15 M DL-Malic acid pH 7.0, 20% w/v Polyethylene glycol 3,350 for Nb70-1F11 Fab-SARS-CoV-2 RBD, 0.2 M Ammonium sulfate, 0.1 M Bis Tris pH 5.5, 25% w/v Polyethylene glycol 3,350 for 1-2C7-SA RBD, 0.2 M Ammonium

formate, pH 6.6, 20% w/v Polyethylene glycol 3,350 for 3-2A2-4-SARS-CoV-2 RBD. Crystals were collected, soaked briefly in 0.2 M Ammonium sulfate, 0.1 M Bis-Tris pH 4.4, 21% w/v Polyethylene glycol 3,350 and 20% glycerol for Nb70-SARS-CoV-1 RBD, 0.15 M DL-Malic acid pH 7.0, 20% w/v Polyethylene glycol 3,350 and 20% glycerol for Nb70-1F11 Fab-SARS-CoV-2 RBD, 0.2 M Ammonium sulfate, 0.1 M Bis-Tris pH 5.5, 25% w/v Polyethylene glycol 3,350 and 20% glycerol for 1-2C7-SA RBD, 0.2 M Ammonium formate, pH 6.6, 20% w/v Polyethylene glycol 3,350 and 20% glycerol for 3-2A2-4-SARS-CoV-2 RBD and were subsequently flash-frozen in liquid nitrogen. Diffraction data were collected at 100 K and a wavelength of 0.987 Å at the BL18U1 beam line for Nb70-SARS-CoV-1 RBD, Nb70-1F11 Fab-SARS-CoV-2 RBD and 1-2C7-SA RBD, 100 K and a wavelength of 1.07180 Å at the BL02U1 beam line for 3-2A2-4-SARS-CoV-2 RBD of the Shanghai Synchrotron Research Facility. Diffraction data were processed using the HKL3000 software (PMID:16855301) and the data-processing statistics are listed in Table S2.

### Structure determination and refinement

The structure was determined using the molecular replacement method with PHASER in the CCP4 suite (PMID: 19461840). Density map improvement by updating and refinement of the atoms was performed with ARP/wARP26 (PMID: 18094467). Subsequent model building and refinement were performed using COOT (PMID: 15572765) and PHENIX (PMID: 12393927), respectively. Final Ramachandran statistics: 96.54% favored, 3.46% allowed and 0.00% outliers for the final Nb70-SARS-CoV-1 RBD structure, 97.28% favored, 2.59% allowed and 0.14% outliers for the final Nb70-1F11 Fab-SARS-CoV-2 RBD structure, 97.15% favored, 2.85% allowed and 0.00% outliers for the final 1-2C7-SA RBD structure and 93.89% favored, 5.79% allowed and 0.32% outliers for the final 3-2A2-4-SARS-CoV-2 RBD structure. The structure refinement statistics are listed in Table S2. All structure figures were generated with ChimeraX and Pymol (PMID: 28158668).

### Conservation analysis of RBD sequences from SARS-CoV-2 and representative sarbecoviruses

SARS-CoV-2 spike sequences collected from December 2019 to June 2022 were obtained from GISAID EpiCoV database. Representative spike sequences from sarbecoviruses were obtained from NCBI Virus database. After filtration, over 8 million SARS-CoV-2 RBD sequences and 46 sarbecovirus RBD sequences were extracted. These RBD sequences were aligned using MAFFT v7.310 program. The entropy was calculated for each residue based on aligned sequences as follows:

$$\text{Entropy} = -\sum_{i=1}^{21} p(x_i) \log(p(x_i)) \tag{1}$$

In which xi are standard amino acids, plus gap. Figures were generated by UCSC ChimeraX 1.3.

### Cryo-EM sample preparation and data collection

The trimeric spike of Omicron BA.1 at concentration of 1.5 mg/mL was incubated with 3-fold molar excess of 3-2A2-4 at room temperature for 1 hours. 4 μL of mixture was applied to glow-discharged holey carbon grids (Quantifoil grid, Cu 300 mesh, R1.2/1.3). The Quantifoil grids were blotted for 2 seconds with filter paper in 100% relative humidity at 8 °C and plunged into the liquid ethane to freeze samples using FEI Vitrobot system (FEI). Images for nanobody-spike trimer complexes were recorded using FEI Titan Krios microscope (Thermo Fisher Scientific) operating at 300 kV with a Gatan K3 Summit direct electron detector (Gatan Inc.) at Tsinghua University. The automated software (AutoEMation2, PMID: 15797731) was used to collect 4795 movies for nanobody-spike trimer complexes in super-resolution mode at a nominal magnification of 81,000× with a pixel size of 1.0979 Å at a defocus range between -1.5 and-1.8 μm. Each movie has a total

accumulate exposure of 50 e-/Å2 fractionated in 32 frames of 175 ms exposure.

## Cryo-EM data processing

Motion Correction (MotionCor2 v.1.2.6, PMID: 28250466), CTF-estimation (GCTF v.1.18, PMID: 26592709), and non-templated particle picking (Gautomatch v.0.56, http://www.mrc-lmb.cam.ac.uk/kzhang/) were automatically executed by TsingTitan.py program. A total of 1,812,056 particles for nanobody-spike trimer complexes were extracted using RELION 3.0.8. Sequential data processing was carried out on cryoSPARC. After 2D classification, the best selected 947,196 particles were applied for initial model and 3D classification.

## Nanobody protection against authentic SARS-CoV-2 infection in K18-hACE2 mice

Animal experiments were conducted in a Biosafety Level 3 (BSL-3) facility in accordance with the National University of Singapore (NUS) Institutional Animal Care and Use Committee (IACUC) (protocol no. R20-0504), and the NUS Institutional Biosafety Committee (IBC) and NUS Medicine BSL-3 Biosafety Committee (BBC) approved SOPs. Eight-week-old female K18-hACE2 transgenic mice (InVivos Ptd Ltd, Lim Chu Kang, Singapore) were used for this study. The mice were housed and acclimatized in an ABSL-3 facility for 72 h prior to the start of the experiment. K18-hACE2 transgenic mice were subjected to pretreatment of nanobody 3-2A2-4 (10 mg/kg) delivered through intraperitoneal injection a day prior to infection. The viral challenge was conducted through intranasal delivery in 25 μL of either $1.7 \times 10^3$ PFU of the infectious SARS-CoV-2 Omicron subvariant BA.1 or $10^3$ PFU of Delta variant. Baseline body weights were measured prior to infection and monitored daily by two personnel post-infection for the duration of the experiment. Mice were euthanized when their body weight fell below 75% of their baseline body weight. To assess the viral load, mice from each experimental group were sacrificed 3 days post inoculation, with lung tissues harvested. Each organ was halved for the plaque assay and histology analysis, respectively. Tissues were homogenized with 0.5 mL DMEM supplemented with antibiotic and antimycotic (Gibco) and titrated in Vero E6 cells using plaque assays. Specifically, for virus titer determination, supernatants from homogenized tissues were serially diluted 10-fold in DMEM supplemented with antibiotic and antimycotic. Of each serial diluted supernatant, 250 μL was added to Vero E6 cells into 12-well plates. After 1 h of incubation for virus adsorption, the inoculum was removed and washed once with PBS. About 1.2% microcrystalline cellulose (MCC)-DMEM supplemented with antibiotic and antimycotic overlay media was added to each well and incubated at 37 °C, 5% $CO_2$ for 72 h for plaque formation. The cells were then fixed in 10% formalin overnight and counterstained with crystal violet. The number of plaques was determined and the virus titers of individual samples were expressed in logarithm of PFU per organ. For histopathological analyses, lung lobes were fixed in 3.7% formaldehyde solution prior to removal from BSL-3 containment. The tissues were routinely processed, embedded in paraffin blocks (Leica Surgipath Paraplast), sectioned at 4 μm thickness, and stained with H&E (Thermo Scientific) following standard histological procedures. For immunohistochemistry, sections were deparaffinized and rehydrated, followed by heat-mediated antigen retrieval, quenching of endogenous peroxidases and protein blocking. Sections were then covered with rabbit anti-SARS-CoV-2 N protein monoclonal antibody (Abcam ab271180, 1:1000 dilution) for 1 h at room temperature. Subsequently, sections were incubated with rabbit-specific HRP polymer (secondary antibody), visualized using chromogenic substrate DAB solution (Abcam), and counterstained with hematoxylin.

## Statical analysis

The technical and independent experiment replicates were indicated in the figure legends. Half-maximal inhibitory concentration (IC50) of nanobodies was calculated by the equation of four-parameter dose inhibition response using Graphpad Prism 8.0. The fold change of the variants relative to D614G in neutralization was calculated by simple division of respective IC50 values. In animal experiments, a two-tailed unpaired Mann-Whitney test was used to assess statistical significance. Statistical calculations were performed in GraphPad Prism 8.0. Differences with p-values less than 0.05 were considered to be statistically significant (**$p < 0.01$).

## Reporting summary

Further information on research design is available in the Nature Portfolio Reporting Summary linked to this article.

## Data availability

All data are available in the main text or the supplementary materials. Source data are provided with this paper. The coordinates and structure factors for the 1-2C7-SARS-CoV-2 Beta-RBD, Nb70-SARS-CoV-2 WT-RBD-P2C-1F11, Nb70-SARS-CoV-1 WT-RBD, 3-2A2-4-SARS-CoV-2 WT-RBD complex have been deposited in Protein Data Bank under accession code 7X2M (https://doi.org/10.2210/pdb7X2M/pdb), 7X2K (https://doi.org/10.2210/pdb7X2K/pdb), 7X2J (https://doi.org/10.2210/pdb7X2M/pdb), 7X2L (https://doi.org/10.2210/pdb7X2L/pdb). The map of SARS-CoV-2 Omicron spike glycoprotein in complex with three nanobody 3-2A2-4 has been deposited in the Electron Microscopy Data Bank under accession codes EMD-33923 (https://www.ebi.ac.uk/emdb/EMD-33923). The nanobody sequences have been deposited in the GenBank under OP612459-OP612496. SARS-CoV-2 spike sequences collected from December 2019 to June 2022 used in this study were obtained from GISAID EpiCoV database (https://www.epicov.org/epi3/frontend#). Representative spike sequences from sarbecoviruses used in this study were obtained from NCBI Virus database (https://www.ncbi.nlm.nih.gov/labs/virus/vssi/#/). Source data are provided with this paper.

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

## Acknowledgements

This study was funded by the National Key Plan for Scientific Research and Development of China (2021YFC0864500, 2022YFC2604103, 2022YFC2303403, 2022YFF1203100 and 2021YFC2300104), the National Natural Science Foundation (92169205, 82150205, 32270983 and 32171202), Wanke Scientific Research Program (20221080056), Tencent Foundation, Shuidi Foundation, TH Capital, Singapore National Medical Research Council Centre Grant Program (CGAug16M009), NUHSRO/2020/066/NUSMedCovid/01/BSL3 Covid Research Work, NUHSRO/2020/050/ RO5 + 5/NUHS-COVID/4, Singapore Ministry of Health MOH-COVID19RF2-0001. We thank the SSRF BL02U1 and BL18U1 beam line for data collection and processing. We thank the X-ray crystallography platform of the Tsinghua University Technology Center for Protein Research for providing the facility support.

## Author contributions

L.Z., X.W., and J.C. conceived and designed the study. M.L., Y.R., Z.A., and B.C. performed most of the experiments with assistance from Y.L., Q.L., J.H., Y.Y., J.W., Y.W., J.C., S.S. J.G., X.S., and Q.Z. B.C., J.D., and Y.C. immunized the alpaca and constructed the yeast library. M.L., Y.L, Q.L., J.H., Y.Y., and J.W. isolated and characterized nanobodies for genetic and phenotypic features. Y.R., J.C., and J.G. solved and analyzed crystal and cryo-EM structures of nanobody bound to RBD or spike trimer. Z.A., Y.W., and J.C. performed the nanobody protection experiment in K18-hACE2 mice. Z.Y. conducted the sequence analysis. L.C. conducted the live SARS-CoV-2 neutralization assay. R.W. constructed the pseudo-viruses of SARS-CoV-2 and variants. M.L. and Y.R. had full access to data in the study, generated figures and tables, and take responsibility for the integrity and accuracy of the data presentation. L.Z., X.W., M.L., and Y.R. wrote the manuscript. All authors reviewed and approved the final version of the manuscript.

## Competing interests

B.C. is an employee of NB BIOLAB Co., Ltd. J.D. is an employee of HplanetBio Co., Ltd. Y.C. is an employee of Hua Bio Co., Ltd. Patent applications have been filed on nanobodies targeting sarbecoviruses. L.Z., M.L., Y.L., J.H., Y.Y., X.S., and Q.Z. are the inventors. The remaining authors declare no competing interests.

## Additional information

[1]Center for Global Health and Infectious Diseases, Comprehensive AIDS Research Center, NexVac Research Center, Department of Basic Medical Sciences, School of Medicine, Tsinghua University, Beijing 100084, China. [2]The Ministry of Education Key Laboratory of Protein Science, Beijing Advanced Innovation Center for Structural Biology, Beijing Frontier Research Center for Biological Structure, Collaborative Innovation Center for Biotherapy, School of Life Sciences, Tsinghua University, Beijing 100084, China. [3]Tsinghua-Peking Center for Life Sciences, Tsinghua University, Beijing 100084, China. [4]Biosafety Level 3 Core Facility, Yong Loo Lin School of Medicine, National University of Singapore, Singapore 119077, Singapore. [5]Laboratory of Molecular RNA Virology and Antiviral Strategies, Department of Microbiology and Immunology, Yong Loo Lin School of Medicine, National University of Singapore, Singapore 119077, Singapore. [6]Infectious Disease Translation Research Programme, Yong Loo Lin School of Medicine, National University of Singapore, Singapore 119077, Singapore. [7]NB BIOLAB Co., Ltd, Chengdu 611137, China. [8]Institute for Hepatology, National Clinical Research Center for Infectious Disease, Shenzhen Third People's Hospital, Shenzhen 518112, China. [9]The Second Affiliated Hospital, School of Medicine, Southern University of Science and Technology, Shenzhen 518112, China. [10]HplanetBio Co., Ltd, Shanghai 200131, China. [11]Hua Bio Co., Ltd, Hangzhou 310018, China. [12]Collaborative and Translation Unit for HFMD, Institute of Molecular and Cell Biology, Agency for Science, Technology and Research, Singapore 138673, Singapore. [13]Institute of Biopharmaceutical and Health Engineering, Tsinghua Shenzhen International Graduate School, Tsinghua University, Shenzhen 518055, China. [14]Institute of Biomedical Health Technology and Engineering, Shenzhen Bay Laboratory, Shenzhen 518132, China. [15]These authors contributed equally: Mingxi Li, Yifei Ren, Zhen Qin Aw, Bo Chen. ✉e-mail: miccjh@nus.edu.sg; xinquanwang@tsinghua.edu.cn; zhanglinqi@tsinghua.edu.cn

