## [Peer Review File · Nature Communications]

Reviewer comments, first round -

Reviewer #1 (Remarks to the Author):

Mingxi et al., describe the identification of a special group of nanobodies from immunized alpaca with breadth of coverage and potency against diverse VOCs including Omicron subvariants and major sarbecoviruses. The authors state that one representative nanobody, 3-2A2-4, discovered a novel and highly conserved epitope located between the cryptic and the outer face of the receptor binding domain (RBD), distinctive from the receptor ACE2 binding site. Cryo-EM and biochemical evaluation suggests that 3-2A2-4 interferes with the structural alteration of RBD required for ACE2 binding. Passive delivery of 3-2A2-4 protected K18-hACE2 mice from infection of authentic SARS-CoV-2 Delta and Omicron. The authors suggest that the identification of these unique nanobodies provides new reference for the development of next generation antibody therapies and design of pan-sarbecovirus vaccines.

Although the study presents some interesting insights there are major concerns on how it constitutes a significant enough conceptual advance. Furthermore, there are different discrepancies between the statements in the main text and the results and some conclusions appear to be overstated.

In the manuscript the most characterized mAb, 3-2A2-4, is indicated as one "representative" nanobody among those generated by the authors. However, 3-2A2-4 seems to be rather an exception in terms of breadth across SARS-CoV-2 related viruses and other sarbecoviruses in comparison to the other nanobodies presented in Figure 2a. This raises questions about the immunogenicity of the site against which 3-2A2-4 is directed and the possibility to use this antigenic site for immunogen design.

The authors also claim that 3-2A2-4 is targeting a novel epitope, but this epitope appears to at least partially overlap with the one of a previously described class 3 anti-RBD monoclonal antibody (i.e. S309) as shown in Figure 3h. Additional epitopes of previously characterized class 3 mAbs might reveal further overlapping with 3-2A2-4 epitope. This raises questions about the aspects of novelty related to this epitope.

Furthermore, the authors suggests that the mechanism of action of 3-2A2-4 is not based on direct competition with ACE2 for binding to RBD, but rather that the nanobody might affect the step before and/or after ACE2 binding (lines 326-327). Indeed, this mechanism of action should be in line with results previously published for class 3 targeting mAbs such as S309. However, in Figure 2a the nanobody 3-2A2-4 was shown to compete with ACE2, and the competition for binding with ACE2 was further reiterated in line 329- 333 while describing the result of the FACS binding to spike trimers expressed on the surface of mammalian cells.

The potency of 3-2A2-4 is also quite difficult to appraise as this nanobody showed an IC50 against the BA.1 authentic virus of 0.360 ug/ml (lines 220-223), whereas the IC50 with the corresponding pseudovirus was 0.032 ug/ml (lines 292-295). Although this is a comparison between multiple rounds of replication and a single entry of virus there is more than a 10-fold difference in terms of neutralizing potency. This difference might be depended on the cell lines used to assess the neutralizing activity.

Additional issues surround the other nanobody characterized in the manuscript Nb70:

As stated by authors in lines 321-323 the premature-S1 shedding and the ensuing transition of the spike protein conformation to post-fusion has been extensively described for different classes of RBD-targeting antibodies (including those directed against class IV antibodies). It is not clear therefore the novelty associated to the characterization of this nanobody.

In lines 307-309 authors state "Nb70 and 3-2A2-4, on the other hand, had no apparent spatial clashes with ACE2, indicating distinct mechanisms from 1-2C7". This sentence is contrast with the SPR data presented in Figure 2a where Nb70 and 3-2A2-4 showed 30-60% and 10-30%

competition with ACE2, respectively.

Minor concerns:

Considering alpaca immunization protocol, which was based only on immunogens representative of SARS-CoV-2 spike or RBD, how do the authors explain the higher neutralizing activity against SARS-CoV-1 for VHH antibodies from clusters b and c than for SARS-CoV-2? Lines 157-158 suggests better epitope exposure in SARS-CoV-1 than SARS-CoV-2.

Line 169: please use "CoV-AbDab database" and provide a link to the database.

Line 167-168: "the 91 cross-neutralizing nanobodies as well as the isolated 124 nanobodies were all dominated by 19-residue long CDR3". Shouldn't the 91 cross-neutralizing nanobodies be among the 124 nanobodies? Please clarify.

A detailed schematic of the protocol used to immunize alpaca should be provided.

In the figure 4e-g, S3 and S4 only the non-linear regression curve is shown. Please provide the symbols for the different concentrations at which the nanobodies were tested.

Lines 315-316: "ACE2 greatly facilitated the formation of proteinase K-resistant core as opposed to the absence of ACE2 (Lane 4 vs. Lane 3)". The total amount of spike protein loaded in the two lanes appears to be different (e.g. lanes 2 and 3).

In lines 323-326 "By contrast, in the presence of 3-2A2-4, the formation of proteinase K-resistant core was substantially reduced regardless of ACE2 (Lane 7 vs. Lane 8) to the levels that indistinguishable from the baseline spike (Lane 3)." The total amount of spike protein loaded in the lanes 7 and 8 appears to be different from the ones in lines 4-6.

Reviewer #2 (Remarks to the Author):

This is a very well written and interesting manuscript describing the isolation of llama-derived VHH antibodies that neutralize SARS-CoV-2 variants. The manuscript provides neutralization data, epitope mapping, several X-ray structures and cryoEM maps, and protection data. The manuscript also provides mechanistic studies to determine how these VHHs neutralize as well as cross-neutralize multiple variants. There are no major issues with the manuscript besides minor typos as well as a few things below:

LOD in Figure 6d and 6g is missing and should be clarified as well as the data points moved to the LOD for the 0 virus values.

Since a different S construct than 2P was used, with a GSAS substitution at the furin cleavage site, the conclusion that 3-2A2-4 caused the trimer to stay closed should be further discussed in relation to other constructs. This is shown in Figure 5. It should also be demonstrated that other particles with the RBD up were not observed.

Add CC1/2 to the X-ray tables to clarify the resolution cutoffs.

Antibody sequences should be provided or deposited to a public database.

Reviewer #3 (Remarks to the Author):

- What are the noteworthy results?

The authors report the identification of a large panel of nanobodies, derived from immunization of llamas with SARS-CoV2 purified proteins and Spike-displaying vaccine particles. A sub-set of these antibodies show superior breadth and potency against a variety of SARS-CoV2 variants and other Sarbecoviruses. The authors present biophysical and structural data for three nanobodies and elucidate the mechanism of neutralization for each. Further, the authors show that the nanobody 3-2A2-4 protects mice from challenge with either Delta or Omicron-BA.1.

- Will the work be of significance to the field and related fields? How does it compare to the established literature? If the work is not original, please provide relevant references.

This work is a valuable contribution to the SARS-CoV2 antibody field. The authors' very comprehensive analysis of a diverse panel of Sarbecoviruses as well as single- and triple-point mutations is thorough. Furthermore, their mechanistic analysis reveals important insights into the multiple ways antibodies can defeat SARS-CoV2.

- Does the work support the conclusions and claims, or is additional evidence needed?
Yes. No major issues are noted.

- Are there any flaws in the data analysis, interpretation and conclusions? - Do these prohibit publication or require revision?

The data analysis, interpretations and conclusions are thorough and sound.

- Is the methodology sound? Does the work meet the expected standards in your field?

Yes.

- Is there enough detail provided in the methods for the work to be reproduced?
Yes, the methods were very detailed.

Minor comments:

Line 232: "...whereas 3-2A2-4 bound to the SARS-CoV-2 wildtype RBD (SARS-CoV-2 WT) at 2.4 Å resolution" should be changed to "... whereas the structure of 3-2A2-4 bound to the SARS-CoV-2 wildtype RBD (SARS-CoV-2 WT) was solved to 2.4 Å resolution".

Figure 3: Use Beta or B.1.351 rather than "SA" to label RBD variant type.

Line 368: remove "as high as" in the statement "... average as high as 796.7 PFU/tissue" as the value indicates the average of the 6 animals rather than a range.

For Figure 4, panels e-g and Figures S2 and S3 only the fits corresponding a representative of three independent experiments are shown. The authors likely display the data in this manner for clarity. In the interest of reproducibility, since it is not possible to assess the agreement between the replicates without error bars shown, can the authors include a Table that details the IC50 values with the standard deviation listed?

Reviewer #1 (Remarks to the Author):

Mingxi et al., describe the identification of a special group of nanobodies from immunized alpaca with breadth of coverage and potency against diverse VOCs including Omicron subvariants and major sarbecoviruses. The authors state that one representative nanobody, 3-2A2-4, discovered a novel and highly conserved epitope located between the cryptic and the outer face of the receptor binding domain (RBD), distinctive from the receptor ACE2 binding site. Cryo-EM and biochemical evaluation suggests that 3-2A2-4 interferes with the structural alteration of RBD required for ACE2 binding. Passive delivery of 3-2A2-4 protected K18-hACE2 mice from infection of authentic SARS-CoV-2 Delta and Omicron. The authors suggest that the identification of these unique nanobodies provides new reference for the development of next generation antibody therapies and design of pan-sarbecovirus vaccines.

Although the study presents some interesting insights there are major concerns on how it constitutes a significant enough conceptual advance. Furthermore, there are different discrepancies between the statements in the main text and the results and some conclusions appear to be overstated.

Response: We appreciate the reviewer's constructive comments on our manuscript, which provide immense guidance for us to improve its overall quality. We have carefully reviewed and responded to your concerns point-by-point in the sections below.

In the manuscript the most characterized mAb, 3-2A2-4, is indicated as one "representative" nanobody among those generated by the authors. However, 3-2A2-4 seems to be rather an exception in terms of breadth across SARS-CoV-2 related viruses and other sarbecoviruses in comparison to the other nanobodies presented in Figure 2a. This raises questions about the immunogenicity of the site against which 3-2A2-4 is directed and the possibility to use this antigenic site for immunogen design.

Response: We apologize for not being clearer about the representability of 3-2A2-4 among the nanobodies tested. What we intended to say was that 3-2A2-4 represented the G3 antibodies where the potency and breadth were the most distinct and outstanding compared to other groups of nanobodies. Indeed, compared to most of the nanobodies in immunized alpaca and many antibodies in naturally infected or vaccinated individuals, antibodies like 3-2A2-4 are not predominant in numbers. However, the sheer existence of these rare nanobodies at least suggests that the conserved epitope is accessible by the nanobodies. Similar extrapolation could also be applied to those rare but broadly neutralizing antibodies found in humans. How to induce or significantly increase the proportion of these rare but broadly neutralizing antibodies are the very task the field has been working on. In this regard, identification of 3-2A2-4 and its rare epitope are at minimum providing an additional site for the next-generation therapeutics and vaccines to target on.

The authors also claim that 3-2A2-4 is targeting a novel epitope, but this epitope appears to at least partially overlap with the one of a previously described class 3 anti-RBD monoclonal antibody (i.e. S309) as shown in Figure 3h. Additional epitopes of previously characterized class 3 mAbs might reveal further overlapping with 3-2A2-4 epitope. This raises questions about the aspects of novelty related to this epitope.

Response: We agree with the reviewer that the 3-2A2-4 epitope is indeed partially overlapped with that of S309, one of the well-characterized class 3 antibodies against SARS-CoV-2/SARS-CoV-1. However, this level of overlap (6 of 17 epitope residues) does not necessarily compromise the unique and novel features of 3-2A2-4.

First, from the broad perspective, the epitope of 3-2A2-4 is shifted substantially to the lower junction between the inner and outer face of RBD compared to that of S309 (Figure 3j). Such shift is not trivial and has led to the profound impact on the unique binding pose and mechanism of action of 3-2A2-4 compared to the published antibodies including S309 (see below).

Second, from the atomic perspective, the binding pose and interacting specificities are considerably different between 3-2A2-4 and S309. Particularly, 3-2A2-4 CDR3 residues F102 and F103 penetrate into a hydrophobic pocket (light blue in the Figure below) and form hydrophobic interactions with residues F338, F342, Y365, V367, L368 and F374 of the RBD core subdomain (Figure **a** below). Hydrogen bonds are also formed with RBD residues G339, N343 and D364. On the other hand, S309 CDR3 binds to the RBD away from the hydrophobic pocket with distinct angle of approach and residue specificity (Figure **b** below). In addition, upon 3-2A2-4 binding, N343 glycan is pushed away from the hydrophobic pocket while in the case of S309 binding the position of N343 remains unchanged (Figure **b** below).

Fig. (a) Close-up view of the interaction between the CDR3 of 3-2A2-4 (in purple) and the hydrophobic pocket residues of RBD (in light blue). **(b)** For comparison with that of 3-2A2-4, the interaction between the CDR3 of S309 (in orange) and RBD is shown. The hydrophobic pocket is colored in light blue. The N343 glycan in RBD bound to 3-2A2-4 is shown as grey sticks while to S309 as yellow sticks.

Third, from the spike trimer perspective, cryo-EM analysis revealed binding of 3-2A2-4 to all three RBDs only in the closed state (Figure 5d). By contrast, S309 binds to the spike trimers with a single open RBD and the closed conformation, both of which have three S309 Fabs bind (Pinto, D. et al., Nature, 2020; Matthew, M. et al., Science, 2022; Zhennan, Z. et al., Nature Communication, 2022). This structural feature suggests that 3-2A2-4 may exert its neutralizing activity through locking or pulling all RBDs into the “down” conformation. This would prevent RBD from transitioning to the “up” conformation required for binding to the receptor ACE2 and downstream fusion steps. This hypothesis is supported by the experiment where 3-2A2-4 was found to substantially inhibit the binding of soluble ACE2 to the surface expressed full-length

spike of SARS-CoV-2. It is also consistent with our biomedical analysis where 3-2A2-4 was found to substantially inhibit the formation of a proteinase K-resistant core, a surrogate maker for the fusion intermediate between viral and cell membrane (Figure 5a). On the other hand, S309 appeared to exert its neutralization through cross-linking, steric hinderance or aggregation of virion (Pinto, D. et al., Nature, 2020).

Furthermore, the authors suggests that the mechanism of action of 3-2A2-4 is not based on direct competition with ACE2 for binding to RBD, but rather that the nanobody might affect the step before and/or after ACE2 binding (lines 326-327). Indeed, this mechanism of action should be in line with results previously published for class 3 targeting mAbs such as S309. However, in Figure 2a the nanobody 3-2A2-4 was shown to compete with ACE2, and the competition for binding with ACE2 was further reiterated in line 329- 333 while describing the result of the FACS binding to spike trimers expressed on the surface of mammalian cells.

Response: We apologize for not being clearer about our description. In Figure 2a, all nanobodies evaluated had a human Fc fragment in the C-terminus connected through a flexible linker residues GGGGS. We suspect that such flexible Fc tail might have partially clashed with ACE2 when analyzed for their competitive binding to SARS-CoV-2 RBD. To test this hypothesis, we performed additional competition experiments using nanobody without the Fc fragment (monomer) and found no competition with ACE2 in binding to SARS-CoV-2 RBD (see below). We have added this new SPR results to Figure S4 in the revised manuscript.

Fig. The competitive binding of (a) dimeric (Fc) or (b) monomeric 3-2A2-4 with ACE2 to SARS-CoV-2 RBD, measured by SPR.

As for the potential mechanism of neutralization, although we stated that 3-2A2-4 might affect the step before and/or after ACE2 binding, we more inclined to the steps before ACE2 binding. This is largely due to the cryo-EM analysis revealing the binding of 3-2A2-4 to all three RBDs only in the closed state (Figure 5d), suggesting that 3-2A2-4 could lock or pull all RBDs into the “down” conformation thereby preventing RBD from transitioning to the “up” conformation required for binding to the receptor ACE2. This hypothesis is supported by the experiment where 3-2A2-4 was found to substantially inhibit the binding of soluble ACE2 to the surface expressed full-length spike of SARS-CoV-2 (Figure 5c). It is also consistent with the ability of 3-2A2-4 to inhibit the formation of a proteinase K-resistant core, a surrogate maker for the fusion intermediate between viral and cell membrane (Figure 5a). In this regard, the mechanism of action of 3-2A2-4 is distinctly different from that of S309 or typical class 3 antibodies where cross-linking, steric hinderance or aggregation of virion were indicated (Pinto, D. et al., Nature, 2020).

The potency of 3-2A2-4 is also quite difficult to appraise as this nanobody showed an IC50 against the BA.1 authentic virus of 0.360 ug/ml (lines 220-223), whereas the IC50 with the corresponding pseudovirus was 0.032 ug/ml (lines 292-295). Although this is a comparison between multiple rounds of replication and a single entry of virus there is more than a 10-fold difference in terms of neutralizing potency. This difference might be depended on the cell lines used to assess the neutralizing activity.

Response: To address this question, we re-evaluated neutralizing activity of 3-2A2-4 against BA.1 pseudovirus and BA.1 authentic live virus. The IC50s remained similar to our original data and reaffirmed that different assay systems indeed offered different absolute IC50 values although the trend was similar. We have updated the results of in the revised manuscript.

Additional issues surround the other nanobody characterized in the manuscript Nb70: As stated by authors in lines 321-323 the premature-S1 shedding and the ensuing transition of the spike protein conformation to post-fusion has been extensively described for different classes of RBD-targeting antibodies (including those directed against class IV antibodies). It is not clear therefore the novelty associated to the characterization of this nanobody. In lines 307-309 authors state “Nb70 and 3-2A2-4, on the other hand, had no apparent spatial clashes with ACE2, indicating distinct mechanisms from 1-2C7”. This sentence is contrast with the SPR data presented in Figure 2a where Nb70 and 3-2A2-4 showed 30-60% and 10-30% competition with ACE2, respectively.

Response: Although Nb70 was inferior to 3-2A2-4 in terms of neutralizing potency and breadth against diverse panel of tested pseudoviruses, we nevertheless decided to choose Nb70 as well as 1-2C7 as reference to highlight the exceptional properties of 3-2A2-4. While the unique structural and neutralizing properties of 3-2A2-4 were being investigated, we also encountered some unexplainable findings where Nb70-Fc, like that 3-2A2-4-Fc, partially competed with ACE2 despite of its epitope was clearly located away from the ACE2 binding site (Figure 2a and Figure 3). Initially, we thought it could be due to the dimeric effect of Nb70-Fc where the flexible linker and long Fc tail could impose some steric clash with ACE2. However, additional experiments conducted during the revision process have unveiled some unexpected insights into the additional mechanism of Nb70, and perhaps some class IV antibodies in interfering with ACE2 binding (see below).

Specifically, we compared the level of competition between each nanobodies (1-2C7, Nb70, and 3-2A2-4 with or without Fc) with soluble ACE2 for binding to RBD, and to see whether the Fc tail was the reason behind competition with ACE2 (see Figure below). As expected, 1-2C7 strongly competed with ACE2, regardless with or without Fc tail (Figure **a** below). 3-2A2-4, on the other hand, only competed with ACE2 when containing the Fc tail, suggesting Fc tail accounted for the partial competition with ACE2 (Figure **b** below). Most surprisingly, Nb70 as well as the typical class IV antibody CR3022 demonstrated partial yet convincing levels of competition with ACE2 irrespective of Fc tail (Figure **c** and **d** below). Similar result on CR3022 competition with ACE2 has also been reported by Hou et al. in Cell Host & Microbe, 2022. These results therefore indicate that Nb70, CR3022, or perhaps some class IV antibodies bind to the inner face of RBD to induce the premature-S1 shedding, which indirectly

affect ACE2 binding to RBD. This would explain why Nb70, with or without Fc, appeared to have similar competition profile with ACE2.

Fig. The competitive binding of dimeric (Fc) or monomeric (a) 1-2C7, (b) 3-2A2-4 and (c) Nb70 with ACE2 to SARS-CoV-2 RBD, measured by SPR. (d) The competition of CR3022 in IgG and Fab form with ACE2 for binding to SARS-CoV-2 RBD was also measured by SPR.

In summary, we have demonstrated distinct mechanisms of three nanobodies (1-2C7, Nb70, and 3-2A2-4) in competing with ACE2. 1-2C7 acts through direct competition for binding to RBD. Nb70 functions indirectly through inducing premature-S1 shedding, whereas 3-2A2-4 acts through locking or pulling RBD into the “down” conformation. In light of these intricate and uniqueness of each nanobodies, it is indeed worthwhile to conduct more thorough analysis for more insights into the mechanism of action.

Minor concerns:

Considering alpaca immunization protocol, which was based only on immunogens representative of SARS-CoV-2 spike or RBD, how do the authors explain the higher neutralizing activity against SARS-CoV-1 for VHH antibodies from clusters b and c than for SARS-CoV-2? Lines 157-158 suggests better epitope exposure in SARS-CoV-1 than SARS-CoV-2.

Response: We were puzzled by the same question when we obtained these results. However, during the revision process, we realized that similar findings have already been reported for human antibodies isolated from recovered COVID-19 patients and Moderna mRNA-1273 vaccinees. For instance, He and colleagues isolated 93 monoclonal antibodies from COVID-19 patients with hybrid immunity and found many presented a better neutralizing ability against SARS-CoV-1 than against SARS-CoV-2 (He, Wanting et al. Nature Immunology, 2022). Another cross-neutralizing antibody BG10-19, isolated from recovered COVID-19 patients, neutralized SARS-CoV-2 with an IC50 of 9 ng/mL while neutralized SARS-CoV-1 with a lower IC50 of 3 ng/mL (Scheid, Johannes F. et al. Cell, 2021). As speculated before, the epitopes recognized

by these antibodies may be better exposed or more accessible in the spike of SARS-CoV-1 than in SARS-CoV-2 due to some delicate changes in epitope residues.

Line 169: please use “CoV-AbDab database” and provide a link to the database.

Response: We have modified this in the revised version. The link to CoV-AbDab database is <http://opig.stats.ox.ac.uk/webapps/covabdab/>.

Line 167-168: “the 91 cross-neutralizing nanobodies as well as the isolated 124 nanobodies were all dominated by 19-residue long CDR3”. Shouldn’t the 91 cross-neutralizing nanobodies be among the 124 nanobodies? Please clarify.

Response: Sorry for the confusion. The 91 cross-neutralizing nanobodies are indeed among the 124 nanobodies. We have modified this as “The 91 cross-neutralizing nanobodies were dominated by 19-residue long CDR3” in the revised version.

A detailed schematic of the protocol used to immunize alpaca should be provided.

Response: We have added an immunization scheme in Figure S1a in the revised manuscript.

In the figure 4e-g, S2 and S3 only the non-linear regression curve is shown. Please provide the symbols for eh different concentrations at which the nanobodies were tested.

Response: As there were too many curves in Figure 4e-g, S2 and S3, we thought it would be clearer if we only displayed the non-linear regression curves without symbols and error bars. In the revised manuscript, as suggested, we provided the complete curves with symbols and bars for Fig.4e-g. For Figures S2 and S3, we included a new Table S1 to show the average IC50 values and related standard deviation for each and every nanobodies presented in Figure 2a.

Lines 315-316: “ACE2 greatly facilitated the formation of proteinase K-resistant core as opposed to the absence of ACE2 (Lane 4 vs. Lane 3)”. The total amount of spike protein loaded in the two lanes appears to be different (e.g., lanes 2 and 3).

Response: At the beginning of our experiment, we added the same amount of spike protein (1µg) into each and every testing tubes. Once the digestion by proteinase K was completed and resultant samples were loaded and analyzed on the gel, the amount of spike protein appeared to be different. In fact, this was what we expected to see given the differential reaction conditions (i.e., trypsin, proteinase K, nanobodies, and etc.) among the different testing tubes. We used an anti-S2 polyclonal antibody to visualize and quantify the amount of proteinase-K resistant core among different testing conditions, as proteinase-K resistant core is exclusively made by the stretch of residues in the S2 domain of the spike protein.

In lines 323-326 “By contrast, in the presence of 3-2A2-4, the formation of proteinase K-resistant core was substantially reduced regardless of ACE2 (Lane 7 vs. Lane 8) to the levels that indistinguishable from the baseline spike (Lane 3).” The total amount of

spike protein loaded in the lanes 7 and 8 appears to be different from the ones in lines 4-6.

Response: Please see our response above. The differences in lanes 7 and 8 compared to those in lanes 4-6 were due to the differential reaction conditions and were in fact what we expected to see.

Reviewer #2 (Remarks to the Author):

This is a very well written and interesting manuscript describing the isolation of llama-derived VHH antibodies that neutralize SARS-CoV-2 variants. The manuscript provides neutralization data, epitope mapping, several X-ray structures and cryoEM maps, and protection data. The manuscript also provides mechanistic studies to determine how these VHHs neutralize as well as cross-neutralize multiple variants. There are no major issues with the manuscript besides minor typos as well as a few things below:

LOD in Figure 6d and 6g is missing and should be clarified as well as the data points moved to the LOD for the 0 virus values.

Response: The viral load in the lung tissue presented in Figure 6d and 6g was measured by the plaque assay. The limit of detection (LOD) was 10 PFU. We have added a line at $y=10$ to indicate the LOD of our assay.

Since a different S construct than 2P was used, with a GSAS substitution at the furin cleavage site, the conclusion that 3-2A2-4 caused the trimer to stay closed should be further discussed in relation to other constructs. This is shown in Figure 5. It should also be demonstrated that other particles with the RBD up were not observed.

Response: The spike protein with the 6P and GSAS substitutions at the furin cleavage site to stabilize the overall structure have been used by several groups for cryo-EM structural studies (PMID: 32075877, 32577660, 32703908, 35120603, 36075908). Particularly, the exact same spike of Omicron BA.1 used in our study containing the 6P and GSAS mutations showed only open conformation with one up-RBD after 3D classification (PMID: 35120603), However, when bound with 3-2A2-4 in our study, all subsets of spikes after 3D classification were in the closed state (FigureS5). We have added this discussion into the revised manuscript.

Add CC1/2 to the X-ray tables to clarify the resolution cutoffs.

Response: We have modified this in the revised version.

Antibody sequences should be provided or deposited to a public database.

Response: As suggested, we have submitted the sequences to GenBank and their accession numbers are indicated below and included in the revised manuscript.

Nanobody	GenBank accession number	Nanobody	GenBank accession number
1H3	OP612459	4-F10-1	OP612478
1-2A1	OP612460	2-2H5-1	OP612479
1-1F11	OP612461	1-2F8	OP612480
2-E5	OP612462	1-2D3-3	OP612481
2-1G11-5	OP612463	2-2F1	OP612482
1-1B10	OP612464	1-1C2	OP612483
1-B1	OP612465	4-2E10	OP612484
1-1E4	OP612466	Nb70	OP612485
1-1E10	OP612467	Nb3	OP612486
1-2B1	OP612468	Nb7	OP612487
1-2C7	OP612469	3-2A2-4	OP612488
1-1G2	OP612470	4-1A10-1	OP612489
1-2D8	OP612471	4-1A10-4	OP612490
1-2F2	OP612472	1-E1-1	OP612491
1-2B3	OP612473	1-C2	OP612492
1-1C7-4	OP612474	1-C3-4	OP612493
1-G1	OP612475	1-2D3-1	OP612494
4-1H1	OP612476	3-2D7-2	OP612495
4-1E1-4	OP612477	1-1G9	OP612496

Reviewer #3 (Remarks to the Author):

- What are the noteworthy results?

The authors report the identification of a large panel of nanobodies, derived from immunization of llamas with SARS-CoV2 purified proteins and Spike-displaying vaccine particles. A sub-set of these antibodies show superior breadth and potency against a variety of SARS-CoV2 variants and other Sarbecoviruses. The authors present biophysical and structural data for three nanobodies and elucidate the mechanism of neutralization for each. Further, the authors show that the nanobody 3-2A2-4 protects mice from challenge with either Delta or Omicron-BA.1.

- Will the work be of significance to the field and related fields? How does it compare to the established literature? If the work is not original, please provide relevant references.

This work is a valuable contribution to the SARS-CoV2 antibody field. The authors' very comprehensive analysis of a diverse panel of Sarbecoviruses as well as single- and triple-point mutations is thorough. Furthermore, their mechanistic analysis reveals important insights into the multiple ways antibodies can defeat SARS-CoV2.

- Does the work support the conclusions and claims, or is additional evidence needed?

Yes. No major issues are noted.

- Are there any flaws in the data analysis, interpretation and conclusions? - Do these prohibit publication or require revision?

The data analysis, interpretations and conclusions are thorough and sound.

- Is the methodology sound? Does the work meet the expected standards in your field?

Yes.

- Is there enough detail provided in the methods for the work to be reproduced?

Yes, the methods were very detailed.

Minor comments:

Line 232: "...whereas 3-2A2-4 bound to the SARS-CoV-2 wildtype RBD (SARS-CoV-2 WT) at 2.4 Å resolution" should be changed to "... whereas the structure of 3-2A2-4 bound to the SARS-CoV-2 wildtype RBD (SARS-CoV-2 WT) was solved to 2.4 Å resolution".

Response: Fixed as suggested. Thanks!

Figure 3: Use Beta or B.1.351 rather than "SA" to label RBD variant type.

Response: We have modified this in the revised version.

Line 368: remove "as high as" in the statement "... average as high as 796.7 PFU/tissue" as the value indicates the average of the 6 animals rather than a range.

Response: Fixed. Thanks!

For Figure 4, panels e-g and Figures S2 and S3 only the fits corresponding a representative of three independent experiments are shown. The authors likely display the data in this manner for clarity. In the interest of reproducibility, since it is not possible to assess the agreement between the replicates without error bars shown, can the authors include a Table that details the IC50 values with the standard deviation listed?

Response: Thanks for your suggestion. As there were too many curves in Figure 4e-g, S2 and S3, we thought it would be clearer if we only displayed the non-linear regression curves without symbols and error bars. In the revised manuscript, we provided the complete curves with symbols and bars for Fig.4e-g. For Figures S2 and S3, we included a new Table S1 to show average IC50 values and related standard deviation for each and every nanobodies presented in Figure 2a.

Reviewer comments, second round -

Reviewer #1 (Remarks to the Author):

In the rebuttal letter the authors have thoroughly addressed all the questions and concerns raised by the reviewer. In addition, the manuscript appears much improved by the inclusion of the new or revised section highlighted in the text.
The reviewer has no further comment.

Reviewer #2 (Remarks to the Author):

All concerns addressed.

Reviewer #1 (Remarks to the Author):

In the rebuttal letter the authors have thoroughly addressed all the questions and concerns raised by the reviewer. In addition, the manuscript appears much improved by the inclusion of the new or revised section highlighted in the text. The reviewer has no further comment.

Response: We appreciate the reviewer's help for the improvement of manuscript.

Reviewer #2 (Remarks to the Author):

All concerns addressed.

Response: Thanks so much!